# Diversity in trap color and height increases species richness of bark and woodboring beetles detected in multiple funnel traps

Jon Sweeney[1]*, Wentao Gao[2], Jerzy M. Gutowski[3], Cory Hughes[1], Troy Kimoto[4], Chantelle Kostanowicz[1], Yan Li[2,5,#a], Chris J. K. MacQuarrie[6], Peter Mayo[1], Qingfan Meng[2,5], Tomasz Mokrzycki[7], Peter Silk[1], Vincent Webster[1,#b], Daniel R. Miller[8]

1 Natural Resources Canada, Canadian Forest Service, Atlantic Forestry Centre, Fredericton, New Brunswick, Canada, 2 Forestry College of Beihua University, Jilin City, Jilin, P. R. China, 3 Department of Natural Forests, Forest Research Institute, Białowieża, Poland, 4 Canadian Food Inspection Agency, Plant Health Surveillance Unit, Burnaby, British Columbia, Canada, 5 Jilin Provincial Key Laboratory of Insect Biodiversity and Ecosystem Function of Changbai Mountains, Jilin City, Jilin, P. R. China, 6 Natural Resources Canada, Canadian Forest Service, Great Lakes Forestry Centre, Sault Ste. Marie, Ontario, Canada, 7 Department of Forest Protection and Ecology, Warsaw University of Life Sciences, Warsaw, Poland, 8 United States of America Department of Agriculture, United States of America Forest Service, Southern Research Station, Athens, Georgia, United States of America

#a Guangxi Eco-Engineering Vocational & Technical College, Liuzhou City, Guangxi, P. R. China
#b Current address: Charter's Settlement, NB, Canada
* jon.sweeney@nrcan-rncan.gc.ca

## Abstract

Early detection of non-native, potentially invasive bark beetles and woodboring beetles (BBWB) (Coleoptera: Buprestidae, Cerambycidae, Disteniidae; Curculionidae: Scolytinae) inadvertently introduced to new habitats via global trade is a critical issue for regulatory agencies in numerous countries. We conducted trapping experiments to evaluate the effects of trap color (black *vs.* green *vs.* purple) and trap height (canopy *vs.* understory) on detection of BBWB in Canada, Poland, USA, and China, using Fluon-treated 12-unit multiple-funnel traps. Each trap was baited with the same pheromone and ethanol lures known to attract several species of BBWB. We predicted BBWB species composition would differ between vertical strata and among trap colors, and that the number of BBWB species detected would increase with greater diversity of trapping methods, i.e., by using more than one color of trap and by placing traps in both the canopy and understory. Our prediction was partially supported, i.e., placing one color of trap in the understory and a different colored trap in the canopy detected more species than did a single trap color placed in either the understory or canopy. However, the combinations of trap height and colors that detected the most species varied among sites. The community of BBWB species captured in traps was significantly affected by trap height and trap color at all sites, with the strongest patterns in the data from Poland and the USA. Black and purple traps caught similar communities of BBWB species in the canopy and understory, but green traps caught a different species assemblage in the canopy in Poland and the

**Data availability statement:** All raw data, metadata, and R statistical computing language code implementing the community analyses are available on the Government of Canada Open Government portal: https://open.canada.ca/data/en/dataset/49898b22-fc9c-495f-ab4a-237d4abe9b37

**Funding:** This project was supported by funds from: the United States Department of Agriculture, Animal and Plant Health Inspection Service, Plant Protection and Quarantine Branch (USDA APHIS-PPQ, https://www.aphis.usda.gov/plant-protection-quarantine) Cooperative Agreement Award 15-8130-0395-CA, received by JS & PS; the Canadian Food Inspection Agency (CFIA, https://inspection.canada.ca/en/about-cfia/science-and-research-cfia) Science Branch Project ID# NRCan-P-1501, received by JS; Natural Resources Canada, Canadian Forest Service, Forest Invasive Alien Species Project, A-base funding (https://natural-resources.canada.ca/our-natural-resources/forests/insects-disturbances/forest-pest-management/responding-invasive-and-alien-forest-pests/13411; received by JS; National Natural Science Foundation of China Youth Fund Project (31600517) https://www.nsfc.gov.cn/english/site_1/index.html, received by QM & YL; and the USDA Forest Service, Forestry Science laboratory (Athens), https://www.fs.usda.gov/organization/Forestry%20Sciences%20Laboratory%20%28Athens%29 received by DRM. The funders played no role in the study design, data collection or analyses, decision to publish, or the preparation of the manuscript.

**Competing interests:** The authors have declared that no competing interests exist.

USA. Effects of trap height and color on species richness were consistent across all four sites within the subfamilies Agrilinae (more species captured in green canopy traps than any other trap height-color combination), Chrysochroinae (more species captured in purple canopy traps than any trap height-color combination) and Scolytinae (more species captured in the understory than the canopy and no effect of trap color), but varied significantly among sites within Cerambycidae subfamilies. None of the species accumulation curves reached an asymptote for any trap color-height combination at any site, indicating that 8–9 traps per site were not sufficient to detect all BBWB species present. Thus, increasing the number of traps deployed per site will increase the BBWB species richness captured and the chances of detecting non-native species that may be present.

## Introduction

Bark beetles and woodboring beetles (Cerambycidae, Buprestidae, Curculionidae: Scolytinae) (BBWB) spend most of their lives under the bark or inside the wood of trees and logs and have been introduced to new continents by human movement of unprocessed logs and untreated solid wood packaging [1–4]. A small fraction of BBWB become established in new habitats where they proliferate, spread, and cause economic and/or environmental harm, i.e., they become invasive forest pests [5,6]. The emerald ash borer, *Agrilus planipennis* Fairmaire (Buprestidae), is a prime example of a woodboring beetle that has become an invasive tree-killing pest in North America, with major economic and ecological impacts [7,8]. Preventing their arrival is the best option. Greenwood et al. [9] present a systems approach for reducing the risks of live insects in wood packaging at each step along the global supply chain, including phytosanitary treatment by heat treatment or fumigation (ISPM 15). However, audits have shown that a small percentage (<0.1%) of wood packaging remains untreated and the sheer volume of shipping containers moving internationally ensures that new introductions of BBWB will continue [10]. The next line of defense is to detect arrivals of BBWB as early as possible. Early detection of small BBWB infestations increases the likelihood of successful containment or eradication of invasive species [11,12].

One method used by regulatory agencies to detect the presence of non-native, potentially invasive BBWB is to survey sites that receive large volumes of imported goods (e.g., ports, industrial parks, and urban forests adjacent to these areas) using flight intercept traps baited with attractive semiochemicals [13–21]. In North America, these surveys have traditionally used black multiple-funnel [22] or black panel traps [23] placed at about head height with the collecting cup 30–50 cm above the ground [15]. These traps work well at detecting species that actively fly in the understory of forests but often fail to detect species that are active in the forest canopy, e.g., *Agrilus* spp. and many species of longhorn beetles [24,25]. Traps in the forest canopy often catch a different assemblage of beetle species than traps in the understory [26–34]. Placing traps in both strata should sample a broader range of BBWB species and potentially increase the number of species detected, including non-native BBWB that may be present.

Trap color also affects the community of BBWB species captured, suggesting that surveys for generic surveillance of BBWB should include traps in more colors than just the standard black [25,35–37]. A number of studies designed to improve detection of emerald ash borer found that purple or green traps were more effective than other colors [38–46]. Subsequent studies showed that green or purple traps were also effective at detecting other species of buprestids [25,47–51] as well as cerambycids [25,51], especially when placed in the upper canopy. Colors attractive to jewel beetles and longhorn beetles differ between species that visit flowers and those that do not [36,37] [e.g., flowering-visiting jewel beetles such as *Anthaxia* spp. favored yellow traps over green whereas non-flower visitors preferred green or purple traps [37]]. Similarly, flower-visiting longhorn beetles preferred yellow, green and blue traps to black traps whereas the reverse was true for non-flower visitors [36].

Previous studies have tested the effects of trap color and/or trap height on the species richness of Buprestidae and Cerambycidae [25], Scolytinae [52], Buprestidae, Cerambycidae, and non-scolytine Curculionidae captured [34], or on individual BBWB species [53–56], but few studies have simultaneously tested the effects of trap color and trap height on the number of species of all three taxa (Buprestidae, Cerambycidae, Scolytinae) captured [51,57]. Our objectives were to evaluate the effects of trap height and trap color on species richness and abundance of BBWB species (Cerambycidae, Buprestidae, Disteniidae, and Curculionidae: Scolytinae) captured in traps, and to determine the combination(s) of trap height and color that captured the greatest number of species per trapping effort, with the overall goal of improving trapping surveillance for detection of non-native BBWB. A trapping system that increases the species richness of target taxa captured should increase the chances of detecting non-native species of related taxa that may be present [20,32]. We also used mean catch per trap of individual species, or their detection rate (proportion of traps that captured at least one specimen), as response variables. Although very few relationships between trap catch and infestation level have been determined for any BBWB species [but see Hanula et al. [58]], mean catch per trap and detection rate provide measures of relative efficacy among trap treatments for detecting a particular species.

We replicated the experiment in mixed deciduous-coniferous forests in Canada, USA, China, and Poland to see whether trends were consistent among sites with different BBWB species assemblages, and to determine the effects of our treatments on detection of a broad range of BBWB species potentially at risk of introduction to our respective countries. Based on the literature and previous experience, we predicted some species would be more common in the canopy (e.g., *Agrilus* spp.) and others more common in the understory (e.g., ambrosia beetles) [24–27,30,31]. Similarly, we predicted that color preferences would differ among taxa [25,36–38], and that green or purple traps would detect more jewel beetles than would black traps. Furthermore, we predicted that the species composition of BBWB would differ between vertical strata and among trap colors, and that the total number of BBWB species captured per trapping effort (number of traps deployed) would be increased by using more than one color of trap and deploying traps in both the canopy and understory.

## Materials and methods

### Experimental design

We conducted a 3 x 2 factorial experiment, testing the effects of three trap colors (black, green, purple), trap height (understory, mid-upper canopy) and their interaction on capture of BBWB at four forested sites in the summer of 2015: 1) Keswick Ridge, New Brunswick, Canada (45.9961 N, 66.8781 W); 2) Georgia, USA, adjacent to the Charlie Elliott Wildlife Center near Mansfield (33.4619 N, 83.7320 W); 3) Białowieża Forest, near Teremiski village, about 8 km NW of Białowieża, Poland (52.7391 N, 23.7686 E); and 4) State-Owned Forest Protection Center of Forestry Experimental Area of Jilin Province, Jiaohe, Jilin Province, P.R. China (127.6990 N, 43.9599 E) (Table 1). The trapping periods (May–September in Canada and China; May–July in USA and Poland) covered the peak flight periods of most species of BBWB at our sites [59–61]. However, we likely missed the peak flight periods of some species of target taxa, e.g., the ambrosia beetles,

**Table 1. Geographical coordinates, dominant tree species and trapping periods of sites where effects of trap height and trap color on detection of bark and wood boring beetles were field tested in 2015.**

| Site | Coordinates | Elevation (m ASL) | Dominant tree species | Trapping period (2015) |
|---|---|---|---|---|
| Keswick Ridge, NB, Canada | 45.9961 N, 66.8781 W | 69 | *Fraxinus americana Populus tremuloides* *Acer saccharum* | 25 May–15 September[1] |
| Mansfield, Georgia, USA | 33.4619 N, 83.7320 W | 207 | *Quercus alba* *Quercus falcata* *Quercus rubra* *Carya tomentosa* *Fagus grandifolia* | 7 May–24 July |
| Białowieża, Poland | 52.7391 N, 23.7686 E | 220 | *Quercus robur* *Picea abies* | 11 May–6 July |
| Jiaohe Forest Farm, Jilin, P.R. China | 127.6990 N, 43.9599 E | 484 | *Quercus mongolica* *Pinus koraiensis* *Fraxinus chinensis Acer mono* | 13 May–7 September |

[1]In New Brunswick, traps were baited with UHR ethanol on 25 May and pheromone lures were added 18 June.

*Xylosandrus germanus* (Blandford) and *Xylosandrus crassisusculus* (Motschulsky), for which peak flights occur in March–April [62]. There were nine replicates per treatment in Georgia (54 traps in total) and eight replicates per treatment (48 traps per site) at the other three sites. Treatments were replicated in randomized complete blocks at all sites with spacing between traps and blocks 12–15 m in Georgia and 25–30 m at the other sites. Trap blocks were set up in the interior of stands to reduce edge effects.

Permission and consent for this research to take place was granted by: the Director General of the Atlantic Forestry Centre, Fredericton, Canada; the Director of the Southern Forestry Center of the US Forest Service, Asheville, North Carolina, USA; the Director of the Forest Research Institute in Sękocin Stary, Poland; the Dean of Forestry College of Beihua University, Jilin, China; and the Director of Jilin Provincial Key Laboratory of Insect Biodiversity and Ecosystem Function of Changbai Mountains, Jilin, China.

### Site descriptions

The site in Keswick Ridge, New Brunswick consisted of two, 15–25 m wide x 500 m long forested strips separated by farm fields and a Christmas tree (*Abies balsamea* (L.) Mill.) plantation. Dominant trees were mature *Fraxinus americana* L., *Populus tremuloides* Michx., and *Acer saccharum* Marsh., with minor components of *Ostrya virginiana* (Mill.) K. Koch, *Pinus strobus* L., *Thuja occidentalis* L., and *A. balsamea*; *Crataegus succulenta* Schrad. ex. Link and *Rubus idaeus* L. were plentiful in the understory. The site in Georgia was relatively flat terrain with mature, closed-canopy, mixed-hardwood forest consisting primarily of *Quercus alba* L., *Q. falcata* Michaux, *Q. rubra* L., *Carya tomentosa* Sargent, and *Fagus grandifolia* Ehrhart with a few scattered *Pinus echinata* Miller. The site in Poland was relatively flat terrain with an oak-lime-hornbeam forest (*Tilio-Carpinetum*) dominated by mature *Quercus robur* L. and *Picea abies* (L.) Karst., with lesser amounts of *Carpinus betulus* L., *Alnus glutinosa* (L.) Gaert., and *Fraxinus excelsior* L., and single scattered *Pinus sylvestris* L., *Betula pendula* Roth, *Populus tremula* L., and *Tilia cordata* Mill.; *Corylus avellana* L., *Sorbus aucuparia* L. and *Frangula alnus* Mill. were present in the understory. The site in Jilin was a mixed broad-leaved/Korean pine forest on a hilltop, dominated by mature *Quercus mongolica* Fisch. Ex Ledeb., *Pinus koraiensis* Siebold & Zucc., *Fraxinus chinensis* Roxb. (*rhynchophylla*), *Acer mono* Maxim., with a minor component of *Acer mandshuricum* Maxim., *Ulmus pumila* L., *Albizia kalkora* (Roxb.) Prain, *Populus davidiana* Dode, *Tilia amurensis* Rupr., and *T. mandshurica* Rupr. & Maxim. In Jilin, traps were set up along a transect that ran along the hilltop, with traps set up along the ridge or on either side of it,

depending on the nearest suitable tree(s) for hanging a trap. Hereafter, sites are abbreviated as follows: Georgia (GA), New Brunswick (NB), Jilin (JI), and Poland (PO).

## Traps and lures

We used 12-unit multiple-funnel traps that were either black, "EAB green" (525 nm, 55%), or "EAB purple" (421 nm, 16.3%; 605 nm, 9.5%; 650 nm, 14.2%) [25,63] treated with a solution of 33% Fluon in water by the supplier (Synergy Semiochemical Corp., Delta, B.C., Canada) to reduce friction and increase catches [64–66]. Collecting cups contained a saturated solution of table salt in water (Canada, China), or 25% propylene glycol in water (ethanol-free RV antifreeze) (USA) or 50% ethylene glycol in water (Poland). A drop of liquid dish detergent was added to reduce surface tension of the trapping solution at all sites except GA. Understory traps were suspended from rope tied between two trees with the collecting cup 30–50 cm above the ground and at least 1 m of horizontal distance between the trap and the adjacent trees. Canopy traps were suspended in the mid to upper crown of dominant or co-dominant trees. The height above ground of canopy traps necessarily varied with the height of the trees in which they were placed, but their relative position was always in the mid to upper tree crown. In NB, JI, and PO, canopy traps were suspended from rope thrown over branches in the mid to upper crown of trees using throwlines, throwbags, and a large slingshot (BigShot®) (SherillTree, Greensboro, NC) following Hughes et al. [67]. In GA, tree climbers placed pulleys and trap lines in the canopy. Height of canopy traps varied according to tree height and ranged from 11–27 m in PO, 18–23 m in GA, 8–16 m in NB, and 4–10 m in JI. We checked and emptied traps every 2–3 weeks, adding trapping solution when required. Beetles in the families Cerambycidae, Buprestidae, Disteniidae, and the Curculionidae subfamily Scolytinae were determined to species whenever possible by using keys and illustrated guides [68–73] (http://www.barkbeetles.info/index.php) and voucher specimens were deposited in respective collections: Atlantic Forestry Centre, Fredericton, NB, Canada; Beihua University, Jilin, PR China; University of Georgia Collection of Arthropods, Athens, Georgia; and the Forest Research Institute, Białowieża, Poland.

Each trap was baited with an ultra high release rate (UHR) lure of ethanol and five separate commercially available pheromone lures attached to the outside of the funnels: 1) racemic 3-hydroxyhexan-2-one; 2) racemic 3-hydroxyoctan-2-one; 3) racemic *syn*-2,3-hexanediol; 4) *E,Z*-6,10-dimethyl-5,9-undecadien-2-ol (*E,Z*-fuscumol); and 5) *E,Z*-6,10-dimethyl-5,9-undecadien-2-yl acetate (*E,Z*-fuscumol acetate) (see Table S1 for source, purity and release rates). These aggregation/sex pheromones are attractive to both sexes of numerous species of longhorned beetles in the subfamilies Cerambycinae, Lamiinae and Spondylidinae [14,74–77]. Ethanol is attractive to many species of ambrosia beetles (Curculionidae: Scolytinae) [78–81] and synergizes attraction of several longhorned beetle species to their pheromones [82–84]. We assumed these lures would have negligible effects on catches of jewel beetles based on previous studies that found no attraction to ethanol [85,86] or to the longhorned beetle pheromones used in our study [24,25]. More recently, Santoiemma et al. [87] found negative effects on mean catches of 6 of 27 species of jewel beetles when UHR ethanol or combinations of ethanol plus cerambycid pheromones were added to unbaited traps; however, the lures had no effect on species richness of jewel beetles captured. At NB, traps were baited with UHR ethanol alone on 25 May and pheromone lures were added 18 June; all other sites used traps baited with ethanol plus pheromone lures for the entire duration of the experiment. The pheromone and ethanol lures were expected to last at least 60 days at 20°C and were not replaced during the experiment except at GA where mean daily high temperatures average 27–32°C in May–July; at GA, the ketol and diol lures were replaced twice (4 weeks and 8 weeks after trap setup) and fuscumol and fuscumol acetate septa were replaced once (6 weeks after trap setup).

## Data analysis

At each site and for each trap, we pooled data across trap check dates to yield one total season catch per trap for each response variable. Response variables included species richness of each family (Buprestidae, Cerambycidae), each relatively abundant subfamily (Agrilinae, Chrysochroinae, Cerambycinae, Lamiinae, Lepturinae, Spondylidinae, Prioninae,

Scolytinae), and all BBWB taxa combined (i.e., Buprestidae + Cerambycidae + Disteniidae + Scolytinae) plus the total catch per trap of individual species for which Cochran's Q test was invalid (see below). If data were missing for a particular treatment and block during a 2–3-week trapping period (e.g., if a trap was found on the ground or damaged during a trap check) we deleted catch data for all treatments in the affected block during the same period from total season catch. Blocks in which zero catch was recorded in all treatments were omitted from analysis.

**Species richness.** We pooled data across sites whenever possible and ran generalized linear mixed-effect models (GLMMs) (PROC GLIMMIX) in SAS 9.4 for Windows (v. 6.2.9200, ©2002–2012, SAS Institute Inc., Cary, NC, USA) testing the effects of trap height, trap color, height*color interaction, and site as fixed factors, with block nested within site as a random factor. To determine whether sites could be pooled, we first tested for site*treatment interactions using a simpler GLMM with treatment (i.e., the six different combinations of trap height and color), site, and site*treatment interaction considered as fixed factors and blocks nested within site as a random factor. This preliminary analysis had two benefits – it indicated whether sites could be pooled to increase statistical power, and it revealed consistency (or lack thereof) among sites in the effects of trap height and color on response variables. These analyses indicated we could pool data from all four sites for species richness of Agrilinae, Chrysochroinae, and Scolytinae, but for all other families and subfamilies, significant site*treatment interactions indicated data needed to be analyzed separately for each site (Table S2). For the latter case, and for mean catch of individual species, we used the model: y = height + color + height*color + block, with blocks random and other factors fixed. We ran models with Gaussian, Poisson and negative binomial distributions and used the corrected Akaike Information Criterion (AICc) to select the model with best fit. We used the Tukey-Kramer post-hoc test adjusted for multiple comparisons (α = 0.05) to compare least square means among trap colors and trap color-height combinations.

**Rate of detecting individual species.** Regulatory agencies are concerned with detecting the presence of non-native species more so than the number of individuals captured. We used Cochran's Q test [88] to determine whether the proportion of traps that captured at least one specimen of a given species at a given site (i.e., rate of detection) was independent of treatment (i.e., varied significantly among the six trap color-height combinations). For Cochran's Q test to be valid, the product of $a \times b$ must be ≥ 24, where $a$ = number of treatments and $b$ = number of blocks in which at least one treatment, but not all treatments, were positive [88], i.e., blocks in which all treatments are negative or all treatments are positive for a given species are not used in the analysis. Thus, species captured in very high numbers, e.g., *Neoclytus mucronatus* (Fabricius), did not qualify for a valid Cochran's Q test because there were too many blocks in which all treatments were positive. For these species, we instead ran generalized linear models as described above, with total catch per trap as the response variable, and used the Tukey-Kramer post-hoc test adjusted for multiple comparisons (α = 0.05) to compare least square means among trap colors and trap color-height combinations.

**Species accumulation curves.** To test our prediction that the total number of BBWB species captured per number of traps deployed would be increased by using more than one color of trap and by deploying traps in both the canopy and understory, we used the iNEXT 4-steps program [89–91] to compare the observed and estimated true BBWB species richness among the six different trap color-trap height treatments, plus all 15 possible pairs of treatments (e.g., green canopy traps plus black understory traps). Each site was analyzed separately. The iNEXT 4-steps program quantifies the sample completeness of survey data across a range of "Hill numbers" ($q$) which give different weight to the relative abundance of species captured. When $q = 1$, all individuals are treated equally, so species are weighted by their relative abundance and the program estimates the number of abundant species present (i.e., Shannon diversity). When $q = 0$, all species are counted equally without regard to their relative abundances, and the program estimates species richness. As recommended for trap samples [91], we first converted raw data on the numbers of each species per trap to incidence-based data (i.e., present = 1, absent = 0), because analyses based on multiple incidence data are less sensitive to clustering of individuals [92]. We report results of the iNEXT 4-steps procedure for $q = 0$ (species richness) rather than $q = 1$ (number of abundant species) because the former is a better measure of potential efficacy of traps for detecting

non-native BBWB species present in low abundance. We used iNEXT 4-steps to estimate: 1) sample completeness (maximum proportion of species present at a site that were captured in the sample); 2) species richness based on sample size, using rarefaction, extrapolation, and asymptotic analysis; and 3) species richness based on non-asymptotic coverage-based rarefaction and extrapolation (the 4th step, evenness of species abundances, was ignored as it was less relevant to our objectives). When sample data are insufficient to infer true species richness (as is often the case), step 3 in iNEXT 4-steps provides coverage-based estimates of species richness for a standardized fraction of individuals, allowing for fair comparisons of estimated species richness among different samples [91]. To examine the general trend in species richness *vs.* increasing diversity in trapping protocol, we averaged estimated species richness in four different categories: 1) one trap color, one height (n = 6); 2) two trap colors, one height (n = 3); 3) one trap color, two heights (n = 6), and 4) two trap colors, two heights (n = 6). We did not test for significant differences in mean species richness among the different categories because (except for our six original treatments of one trap color at one height) the data were not independent. We also ran iNEXT 4-steps on the entire set of trap samples from each site (i.e., 54 traps in GA, 48 traps in JI, NB, and PO) to compare estimated BBWB species richness among sites.

**Community composition.** The community of beetles caught in the three trap colors and two trap positions was analyzed to determine if trap position or trap color influenced the species composition captured. We analyzed the species matrix separately for each site. First we performed nonmetric multidimensional scaling (NMDS) [93] with the counts of each species as the raw data, for our initial conditions in the NMDS we assumed $k = 2$ dimensions, 20 random starts and up to 200 iterations, using the Bray-Curtis distance measure. Our final stress values were 0.248, 0.271, 0.270, and 0.192 for GA, JI, NB, and PO, respectively. The NMDS assigns the community collected in each trap to a position in ordination space; communities that are more like each other are located closer together in the plane. We then assigned each community in ordination space to its treatment groups (trap position and trap color) and determined if the groupings of communities were more similar using a permutational multivariate analysis of variance (perMANOVA) [94] with the community data as the response variable and the trap position and trap color as the predictor variables. The perMANOVA tests the null hypothesis that the location of centroid of each group of communities identified by the predictor variables are located in the same position in ordination space. This test is predicated on the assumption that all the community groups assigned to each level of each predictor variable have similar variances around the group centroid (i.e., similar dispersion). This assumption, referred to as the multivariate homogeneity of group dispersions (i.e., equal variance among the levels of the predictor variables) [95], can be tested by computing the beta dispersion and assessing the differences in beta dispersion among each level of the predictor variables both graphically and by ANOVA. A non-significant result of the ANOVA indicates that the dispersion of the communities for each level of the given predictor are homogenous. Thus, if the results of the perMANOVA are significant we can assume that this result occurs because the group centroids are in different locations and not caused by unequal dispersion. When the results of the dispersion test are significant it means that the communities in a given trap height or trap color are more different than the other communities collected in the other trap height and trap color combinations and that the results of the perMANOVA should be interpreted with caution. In the perMANOVA, as in the previous analyses, we modelled trap color and trap height as interacting main effects. Similarly, since the traps were arranged in a replicated blocked design, we also included a blocking effect in the specification of the perMANOVA to control for inter site variability in community structure (akin to the random site effect in the previous analyses). When there was a significant main effect of trap position or color in the perMANOVA we performed a pairwise perMANOVA to test for differences among levels of the main effects (999 permutations with Holm's correction for multiple comparisons). We did not test for differences in levels if we determined a significant interaction between position and color. NMDS, perMANOVA, and assessment of multivariate dispersion were done in the R statistical computing environment [96] using functions in the vegan package [97]. Functions in the RVAideMemoire [98] and ecodist [99] packages were used to calculate the pairwise perMANOVA.

Finally, we used blocked indicator species analysis (ISA) [100] to determine species indicative of a particular trap height, trap color, and trap height-color combination. These analyses were done in PCORD [101,102], separately for each site. Indicator values (IV) range between 0 and 100; the larger the IV value for a species and grouping (e.g., green-canopy traps), the greater the relative abundance and constancy of that species within that grouping. Constancy is the proportion of replicates of a given trap height-color combination that captured at least one specimen of a given species. PCORD evaluates the statistical significance of IV values by Monte Carlo randomization, randomly reassigning sample units to treatment groups, repeating the IV calculation, iterating these steps 4999 times, and then calculating the proportion of permutations in which the randomized maximum IV for each species was greater than or equal to the observed maximum IV. Small *P*-values indicate species that are more abundant and constant (i.e., more prevalent) in a particular trap height-color combination than would be expected by chance. The ISA in PCORD does not correct the *p* values for multiple comparisons and instead reports the number of indicator species that would be expected by chance for case-wise alpha levels of 0.05, 0.01, and 0.001. We used a case-wise alpha of $P \leq 0.001$ because this ensured that fewer than one species would be considered an indicator purely by chance.

All raw data, metadata, and R statistical computing language code implementing the community analyses are available on the Government of Canada Open Government portal [103].

## Results

We captured 339 species (43 Buprestidae, 204 Cerambycidae, 2 Disteniidae, 90 Scolytinae) and 37,095 specimens (1,369 Buprestidae, 28,329 Cerambycidae, 25 Disteniidae, 8,736 Scolytinae) of target taxa in total (Table 2; Table S3).

**Table 2. Numbers of species and specimens of Cerambycidae, Buprestidae, Curculionidae: Scolytinae, and Disteniidae captured in color x trap height factorial experiment conducted in 2015 in: Georgia, USA; Jilin, China; New Brunswick, Canada; and the Białowiez˙a Forest, Poland.**

| Family | Site | species | specimens | singletons | doubletons |
|---|---|---|---|---|---|
| Buprestidae | Georgia | 17 | 259 | 4 | 2 |
| | Jilin | 10 | 78 | 1 | 2 |
| | New Brunswick | 5 | 212 | 1 | 0 |
| | Poland | 11 | 807 | 2 | 0 |
| | **total** | **43** | **1,368** | **8** | **4** |
| Cerambycidae | Georgia | 72 | 13,676 | 13 | 9 |
| | Jilin | 60 | 8,664 | 15 | 7 |
| | New Brunswick | 36 | 723 | 7 | 4 |
| | Poland | 51 | 3,915 | 12 | 3 |
| | **total**[1] | **204** | **26,978** | **47** | **23** |
| Curculionidae: Scolytinae | Georgia | 46 | 4,173 | 9 | 6 |
| | Jilin | 11 | 1,323 | 6 | 0 |
| | New Brunswick | 19 | 1,413 | 4 | 1 |
| | Poland | 25 | 1,827 | 4 | 2 |
| | **total**[1] | **90** | **8,736** | **23** | **9** |
| Disteniidae | Georgia | 1 | 20 | 0 | 0 |
| | Jilin | 1 | 5 | 0 | 0 |
| | **tota**l | **2** | **25** | **0** | **0** |
| **Target taxa** | **Grand total** | **339** | **37,095** | **78** | **36** |

[1]Overall number of species detected per family was less than sum of species detected per family per site when sites were pooled because 22 species (15 Cerambycidae, 7 Scolytinae) were captured at two sites and 2 species (Scolytinae) were captured at three sites.

The most species-rich and abundant family was the Cerambycidae, comprising 60% of species and 73% of specimens, followed by Scolytinae with 26% of species and 24% of specimens, and Buprestidae with 13% of species and less than 4% of specimens. About one third of all target species captured consisted of singletons (78 species, 23%) and doubletons (36 species, 11%) (Table 2), similar to results of other BBWB trapping studies [e.g., 60,61]. Eight species (1 cerambycid, 7 scolytines) were captured in countries where they were not native (Table S3).

More than half of all target taxa (202 of 339 species, 60%) were captured in both the canopy and understory, but 71 (21%) and 66 species (20%) were captured exclusively in canopy and understory traps, respectively (Fig 1A). Canopy traps captured more buprestid species than did understory traps, with 15 of 43 species (35%) captured only in the canopy compared to 4 species (9%) captured only in the understory (Fig 1A). When singletons and doubletons were excluded, the relative performance of canopy traps for detecting buprestids was even clearer, with 8 of 31 species (26%) captured only in the canopy and 0 captured only in the understory. In contrast, understory traps favored scolytine detection, with 25 of 90 species (28%) captured only in the understory compared to 16 species (18%) captured only in the canopy (Fig 1A). When singletons and doubletons were excluded, 10 of 64 scolytine species (16%) were trapped exclusively in the understory compared to 5 species (8%) trapped exclusively in the canopy (Fig 1A). Similar numbers of Cerambycidae (including Disteniidae) species were captured exclusively in either the canopy or the understory, whether singletons and doubletons were included (Canopy: 40 species; Understory: 37 species) or excluded (Canopy: 11 species; Understory: 8 species) (Fig 1A).

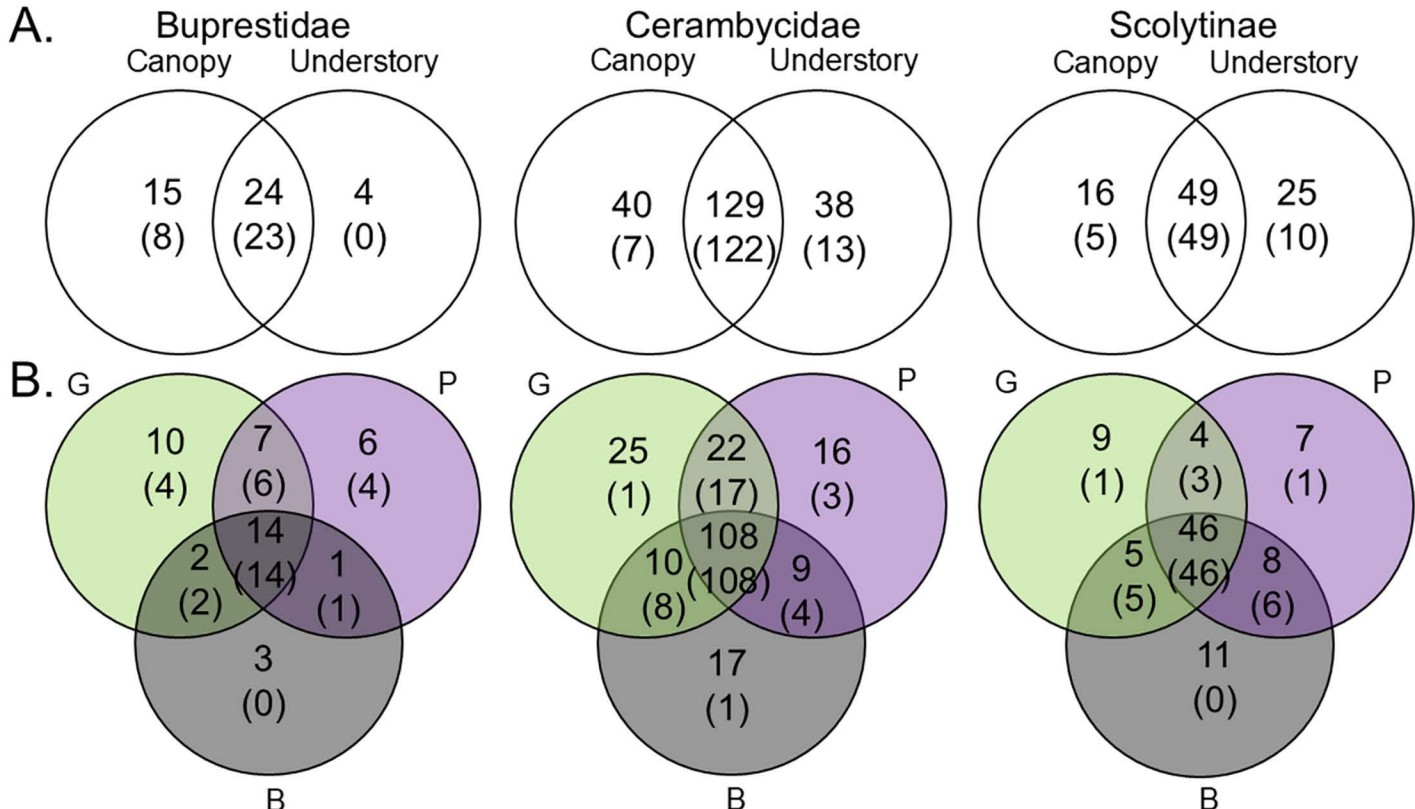

**Fig 1. Numbers of species of Buprestidae, Cerambycidae, and Scolytinae captured in traps that were: A) placed in the Canopy *vs.* Understory; and B) Green (G), Purple (P), or Black (B) in color. For simplicity of illustration, the two Disteniidae species are included within the Cerambycidae. Numbers in parentheses exclude singletons and doubletons.**

About half of all target species captured (167 of 339, 49%) were captured in all three colors of traps, with just 29–34 species (9–10%) captured exclusively in either black, green or purple traps (Fig 1B). The percentage of species captured exclusively by any single trap color ranged from 8–12% for cerambycids and scolytines. When singletons and doubletons were excluded, 2% or less of cerambycids (1–3 of 139 species) or scolytines (0–1 of 62 species) were captured exclusively by one trap color (Fig 1B). However, green and purple traps captured more buprestid species than did black traps, with 10 (23%) and 6 (14%) species captured only in green traps and purple traps, respectively, compared to only 3 species (7%) captured exclusively in black traps (Fig 1B). When singletons and doubletons were excluded, 4 of 31 buprestid species (13%) were captured exclusively in green traps, 4 species (13%) were captured exclusively in purple traps, and none were captured in black traps (Fig 1B). All but 6 of 49 specimens of these seven species were captured in green or purple traps (Table S3).

## Species richness

Species richness of Buprestidae captured in traps was significantly affected by trap height in GA and PO, trap color in NB and PO, and the interaction between trap height and color in GA (Fig 2A–D; Table S4). Green canopy traps captured the most buprestid species on average at every site, but differences were significant only in GA (green canopy > green understory) and PO (green canopy traps > all other treatments). Results were clearer and more consistent among sites when analyzed within subfamily. For both Agrilinae and Chrysochroinae, site*treatment interactions were not significant (Table S2) so data from all sites were pooled. More species of Agrilinae were captured in green canopy traps than any other trap color-height combination, followed by green understory traps, with fewest species captured in black or purple-understory traps (Fig 2E, Table S4). Conversely, more species of Chrysochroinae were captured in purple canopy traps than any other trap color-height combination (Fig 2F, Table S4).

The effects of trap height and color on the number of cerambycid species captured varied among sites (Table S2). The average number of cerambycid species captured per trap was significantly affected by trap height in GA and PO, by trap color in JI, NB and PO, and by the interaction between trap height and color in GA and PO (Fig 3A–D, Table S4). In GA, green understory traps captured more cerambycid species than black canopy traps (Fig 3A) whereas in JI, green traps captured fewer species than black understory traps or purple traps in either the understory or canopy (Fig 3B). Green traps captured the most cerambycid species in NB but means did not separate (Fig 3C) and in PO, green canopy traps captured significantly more cerambycid species than any other trap color-height combination except purple canopy traps (Fig 3D).

Effects of trap height and color on species richness of cerambycids varied among sites even when analyzed within subfamilies (Tables S2 and S4). Species richness of Cerambycinae was unaffected by trap height or color in GA and NB (Figs. 4A,C) but significantly affected by trap color in JI (Fig 4B, black and purple > green) and by trap height in PO (Fig 4D, canopy > understory). Species richness of Lamiinae was unaffected by trap height or trap color in GA (Fig 5A) but effects of trap height were significant in JI (Fig 5B) and PO (Fig 5D) and nearly significant ($P = 0.06$) in NB (Fig 5C), with greater mean captures in canopy traps *vs.* understory traps (Table S4). Lamiinae species richness in PO was also affected by trap color and the interaction between trap color and height, with greater captures in green canopy traps than all other combinations except purple canopy traps (Fig 5D). Lepturine species richness was significantly affected by trap height in GA only, with more species captured in the understory than the canopy (Fig 6A; Table S4). Trap color significantly affected lepturine species richness in JI (Fig 6B), NB (Fig 6C), and PO (Fig 6D) but the color that captured the most species varied among sites. Purple and black traps captured more lepturine species than green traps did in JI (Fig 6B) whereas green traps captured more species than purple traps did in NB (Fig 6C) and more species than black traps did in PO (Fig 6D). Only five prionine species were captured, four in GA, one in JI (*Prionus insularis* Motschulsky), and none in NB or PO (Table S4). Species richness of Prioninae was significantly affected by trap color in GA (with more species captured in black or purple traps than in green traps) (Fig 7A), and by trap height in JI, with more captures of *P. insularis* in

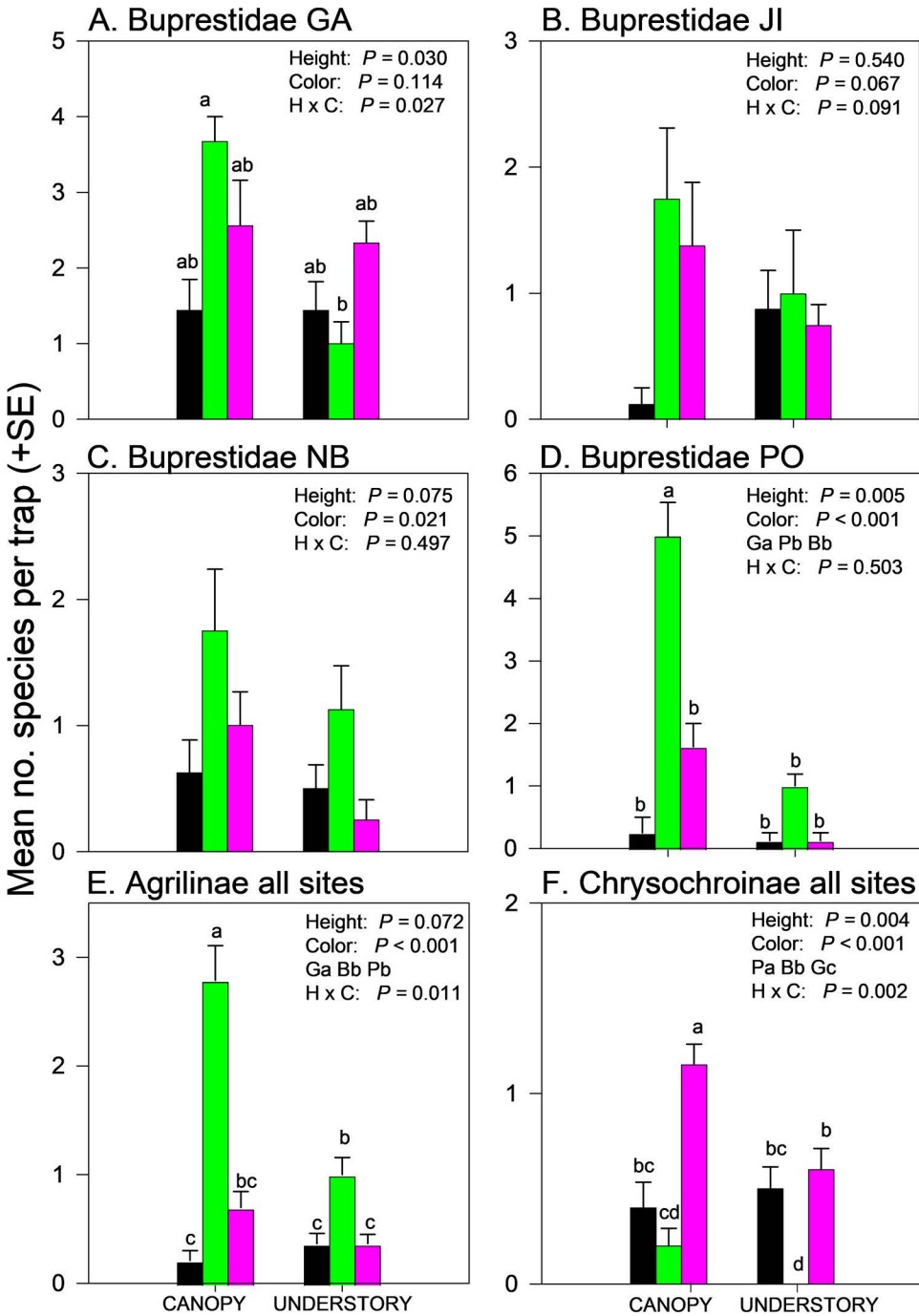

**Fig 2. Effect of trap color (Black, Green, Purple) and trap height (Canopy, Understory) and their interaction on mean ( ±SE) number of species of jewel beetles captured in multiple funnel traps: A) Buprestidae Georgia, USA; B) Buprestidae Jilin, China; C) Buprestidae New Brunswick, Canada; D) Buprestidae Białowiez˙a, Poland; E) Agrilinae, all sites pooled F) Chrysochroinae, all sites pooled. Data were pooled among sites when the interaction between site and treatment was not significant (P>0.05). Different lowercase letters associated with means and color abbreviations (i.e., B, G, P) indicate significant differences (Tukey-Kramer test on least square means, P≤0.05).**

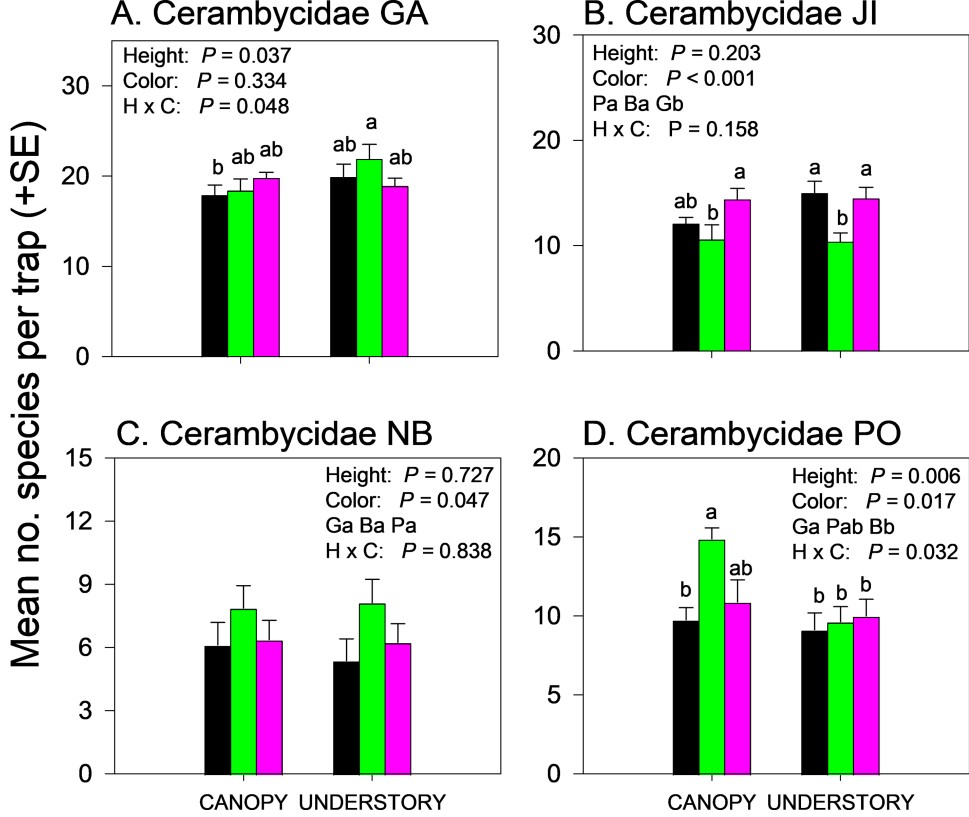

**Fig 3. Effect of trap color (black, green, purple) and trap height (canopy, understory) and their interaction on mean ( ±SE) number of species of Cerambycidae captured in multiple funnel traps in: A) Georgia, USA; B) Jilin, China; C) New Brunswick, Canada; and D) Białowieẑa, Poland.** Different lowercase letters associated with means and color abbreviations (i.e., B, G, P) indicate significant differences (Tukey-Kramer test on least square means, *P* ≤ 0.05).

the understory than the canopy (Fig 7B). Only six species of Spondylidinae were captured, three species in each of JI and PO, one of which was shared between the sites (*Tetropium castaneum* (L.)), and a single specimen of *Tetropium cinnamopterum* Kirby in NB (Table S4). Spondylidinae species richness was not affected by trap height or color in JI (Fig 7C) but was affected by trap height in PO, with more species captured in the understory than the canopy (Fig 7D).

In contrast to the Cerambycidae subfamilies, the effects of trap height and color on species richness of Scolytinae captured was similar at all four sites (Table S2), allowing data to be pooled. Scolytinae species richness was greater in understory traps than canopy traps but trap color and the interaction between trap height and color had no effect (Fig 8A) (Table S4).

Effects of trap color and height on species richness of total target taxa captured varied among sites (Fig 8B–E; Table S2), with a significant effect of trap height in GA (*P* = 0.004, Fig 8B, understory > canopy), significant effects of trap color (*P* = 0.004), trap height (*P* = 0.044), and height-color interaction (*P* = 0.034) in JI (Fig 8C; Table S2), no effects of trap color or height in NB (Fig 8D), and a significant effect of trap color in PO (*P* = 0.004, Fig 8E; Table S4). In JI, black understory traps captured more species of total target taxa than did black canopy traps, or green traps in the canopy or understory; numbers of species captured in purple traps were intermediate (Fig 8C). In PO, the effects of trap height (*P* = 0.058) and height-color interaction (*P* = 0.060) were nearly significant, with more species of total target taxa captured in green canopy traps than in green understory traps or in black traps in either the canopy or understory (Fig 8E; Table S4).

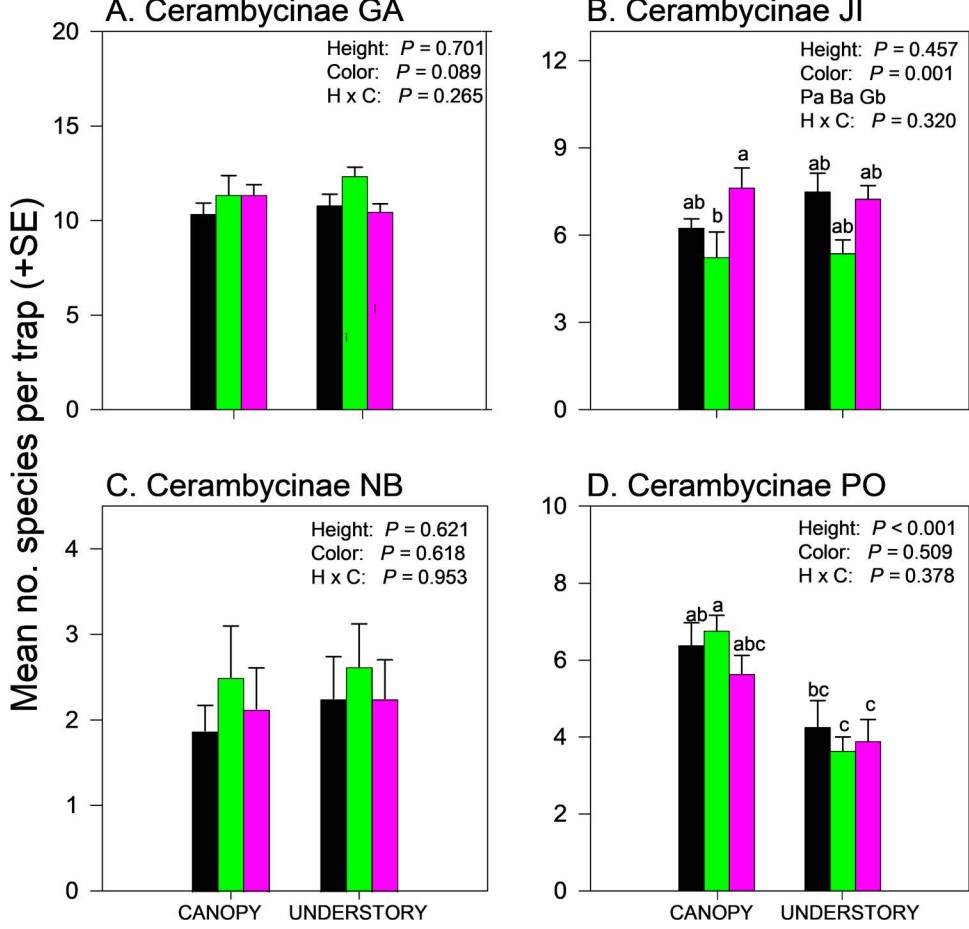

**Fig 4. Effect of trap color (black, green, purple) and trap height (canopy, understory) and their interaction on mean ( ±SE) number of species of longhorn beetles in the subfamily Cerambycinae captured in multiple funnel traps in: A) Georgia, USA; B) Jilin, China; C) New Brunswick, Canada; and D) Białowiez`a, Poland. Different lowercase letters associated with means and color abbreviations (i.e., B, G, P) indicate significant differences (Tukey-Kramer test on least square means, *P* ≤ 0.05).**

### Rate of detecting individual species

Of the 172 species (23 Buprestidae, 103 Cerambycidae, 1 Disteniidae, 45 Scolytinae) for which Cochran's Q test was valid, the rate of detecting at least one specimen differed significantly among trap color-height combinations for 62 species (36%) (14 Buprestidae (61%), 35 Cerambycidae (34%), and 13 Scolytinae (29%)) (Table S3). Except for *Agrilus sulcicollis* Lacordaire, which was captured mainly in purple canopy traps, most species in the Agrilinae subfamily were captured in green traps, and mainly in the canopy (Table S3). Among the cerambycids for which detection rate was not independent of trap treatment, some species were captured more frequently in canopy traps, e.g., *Clytus tropicus* (Panzer), *Neoclytus jouteli* Davis, *Xylotrechus antilope* (Schoenherr), *Parelaphidion aspersum* (Haldeman), *Saperda scalaris* (L.), and *Sphenostethus taslei* (Buquet), and others were captured more frequently in understory traps, e.g., *Elaphidion mucronatum* (Say), *P. insularis*, *Xylotrechus colonus* (Fabricius), *Tetropium fuscum* (Fabricius), and *T. castaneum*. Among the 13 species of Scolytinae for which rate of detection was not independent of trap treatment, 10 were captured more frequently in the understory, e.g., *Anisandrus dispar* (Fabricius), *Xyleborinus attenuatus* (Blandford), *Xylosandrus crassiusculus* (Motschulsky), and 3 [*Scolytus carpini* (Ratzeburg), *Pityogenes chalcographus* (L.), and *Xyleborus monographus* (Fabricius)] were captured more frequently in the canopy (Table S3).

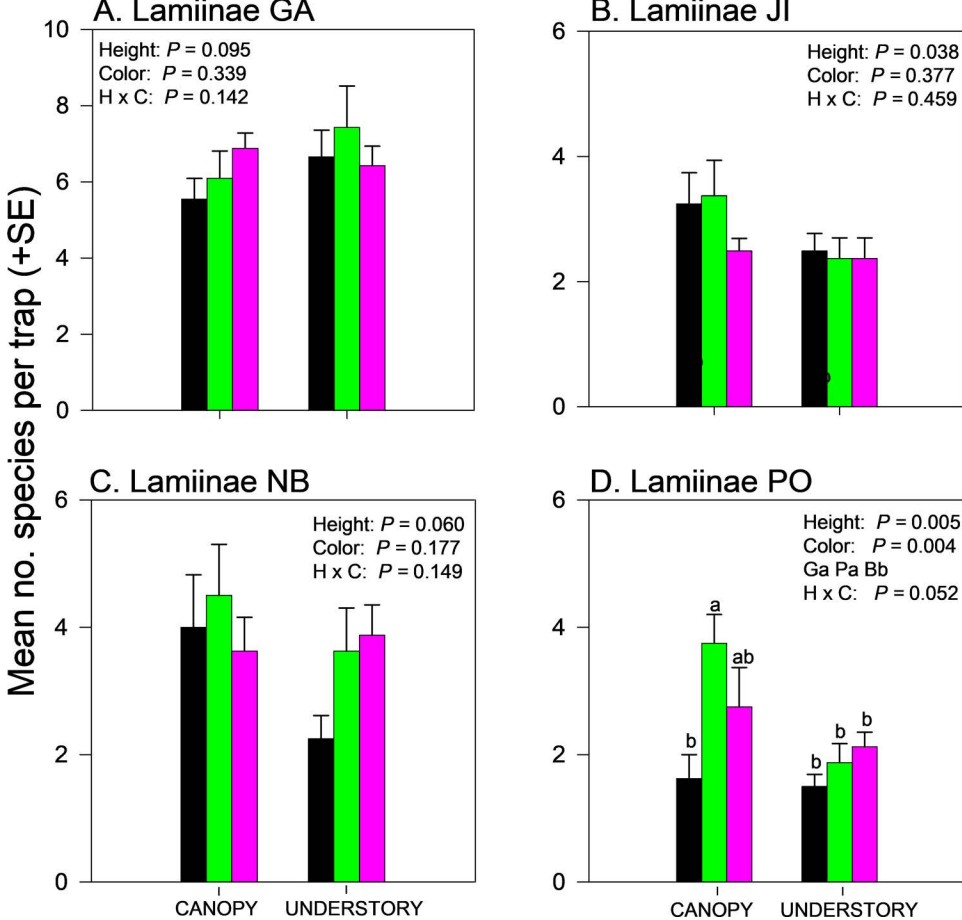

**Fig 5. Effect of trap color (black, green, purple) and trap height (canopy, understory) and their interaction on mean ( ±SE) number of species of longhorn beetles in the subfamily Lamiinae captured in multiple funnel traps in: A) Georgia, USA; B) Jilin, China; C) New Brunswick, Canada; and D) Białowieźa, Poland. Different lowercase letters associated with means and color abbreviations (i.e., B, G, P) indicate significant differences (Tukey-Kramer test on least square means, $P \leq 0.05$).**

There were 14 species (11 Cerambycidae, 3 Scolytinae) captured too frequently for a valid Cochran's Q test that were instead analyzed by GLMMs. Trap height significantly affected mean catch of eight cerambycid species with greatest catches in the canopy for five species [*Phymatodes testaceus* (L.), *Plagionotus detritus* (L.), *P. pulcher* (Blessig), *Anelaphus pumilus* (Newman), *Ecyrus dasycerus* (Say)] and greatest catches in the understory for three species [*Neoclytus acuminatus* (Fabricius), *N. mucronatus* (Fabricius), *Graphisurus fasciatus* (DeGeer)] (Table S5). Trap color significantly affected mean catch of ten cerambycid species, with black or purple traps preferred to green traps for six species and green traps preferred to black traps for two species (Table S5). Trap height significantly affected mean catch of two scolytine species. *Cnestus mutilatus* (Blandford) was more abundant in the canopy than the understory, while the reverse was true for *X. crassiusculus*; catch of the latter species was also significantly greater in black traps than in green traps (Table S5).

## Species accumulation curves

None of the species accumulation curves (showing the total number of BBWB species captured per number of traps deployed) reached an asymptote for any single trap color-height combination at any site (Fig 9A–D), indicating that

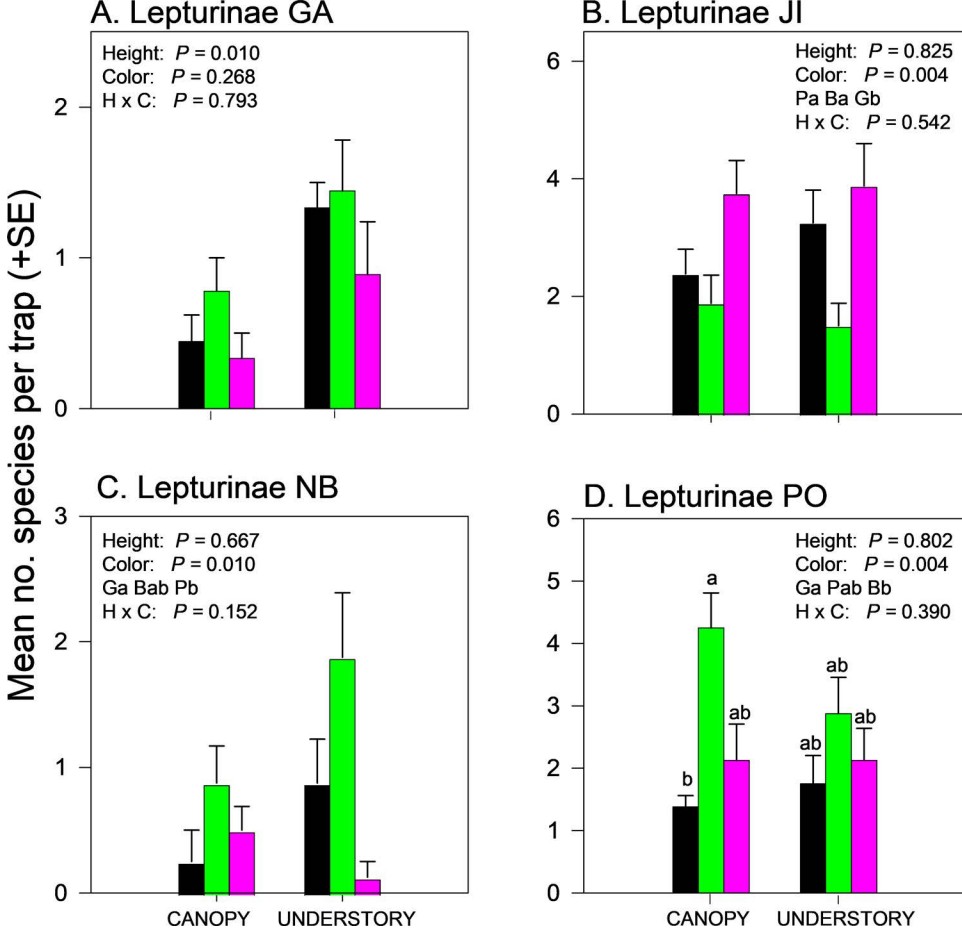

**Fig 6. Effect of trap color (black, green, purple) and trap height (canopy, understory) and their interaction on mean ( ±SE) number of species of longhorn beetles in the subfamily Lepturinae captured in multiple funnel traps in: A) Georgia, USA; B) Jilin, China; C) New Brunswick, Canada; and D) Białowiez˙a, Poland. Different lowercase letters associated with means and color abbreviations (i.e., Ba, Gb, Pb) indicate significant differences (Tukey-Kramer test on least square means, *P* ≤ 0.05).**

8–9 traps per site were not sufficient to detect all BBWB species present. Estimates of sample completeness indicated that a minimum of 14–75% of BBWB species were not detected in samples of 8 (JI, NB, PO) or 9 traps (GA) of a single color placed at one height, and a minimum of 8–49% of species were not detected in samples of 16 (JI, NB, PO) or 18 traps (GA) using combinations of two different trap color-height treatments (Table 3; Figs S1–S4). Species richness also failed to reach an asymptote when data from the entire sample of 48–54 traps per site were analyzed (Table S6, Fig S5). Sample completeness at q = 0 ranged from 0.74–0.84 indicating that at least 16–26% of BBWB species present were not detected (Table S6). The observed (Ob) and asymptotic estimates (AE) of true species richness (95% confidence interval) were highest for GA (Ob: 136; AE: 162 (139–185), lowest for NB (Ob: 60; AE: 81 (53–108), and intermediate for JI (Ob: 82; AE: 102 (82–122) and PO (Ob: 87; AE: 107 (84–129) (Table S6). Coverage-based estimates of BBWB species richness standardized for a maximum coverage of 0.988 confirmed that GA had greater relative BBWB species diversity than the other sites (GA: 153; PO: 99; JI: 94; and NB: 70) (Table S6).

Our prediction that diversifying trap color and trap position would increase the number of BBWB species detected was partially supported. There was a general trend for increased mean species richness with increased diversity in trap color and trap height (Table 3). Depending on the site, using one color of trap in the understory and a different colored trap in

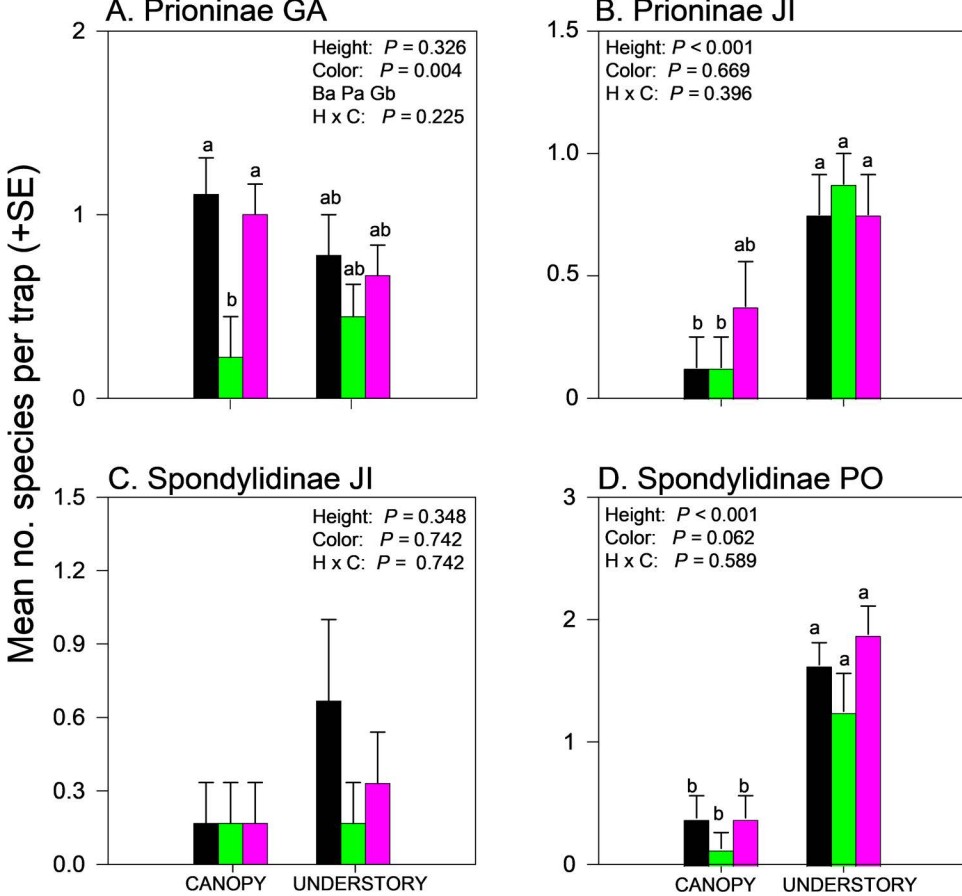

**Fig 7. Effect of trap color (black, green, purple) and trap height (canopy, understory) and their interaction on mean ( ±SE) number of species of longhorn beetles captured in multiple funnel traps in the subfamilies: A) Prioninae, Georgia, USA; B) Prioninae, Jilin, China (JI); C) Spondylidinae, JI; and D) Spondylidinae, Białowieżˈa, Poland. Different lowercase letters associated with means and color abbreviations (i.e., B, G, P) indicate significant differences (Tukey-Kramer test on least square means, $P \leq 0.05$).**

the canopy (i.e., 2 colors x 2 heights) detected 1–7 more (observed/rarefied) species in the same number of traps over the course of the trapping season than did a single trap color placed in either the understory or canopy (Table 3). The average coverage-based estimates (CBE) of true species richness were also greater (by 6–13 species) when two colors and trap heights were deployed *vs.* only one, except in JI, where purple-understory traps had the highest CBE (Table 3). However, the combination of trap colors and heights that detected the most BBWB species varied among sites (Table 3). For example, in PO, black understory traps detected 22 fewer species than did a combination of purple understory and green canopy traps, whereas in NB, black canopy traps detected as many species as the combination of black canopy and green understory traps, and more species than any other combination (Table 3).

## Community composition

We detected non-equal dispersion in the effect of trap color in the data from PO and JI. At both sites the difference in dispersion was small, and always between black traps and green traps, with dispersion of the communities in the black traps being less than that in the green traps. This result indicates the results of the perMANOVA for both sites should be interpreted with caution. There was a significant effect of trap color and trap position at all trapping locations (Table 4). In

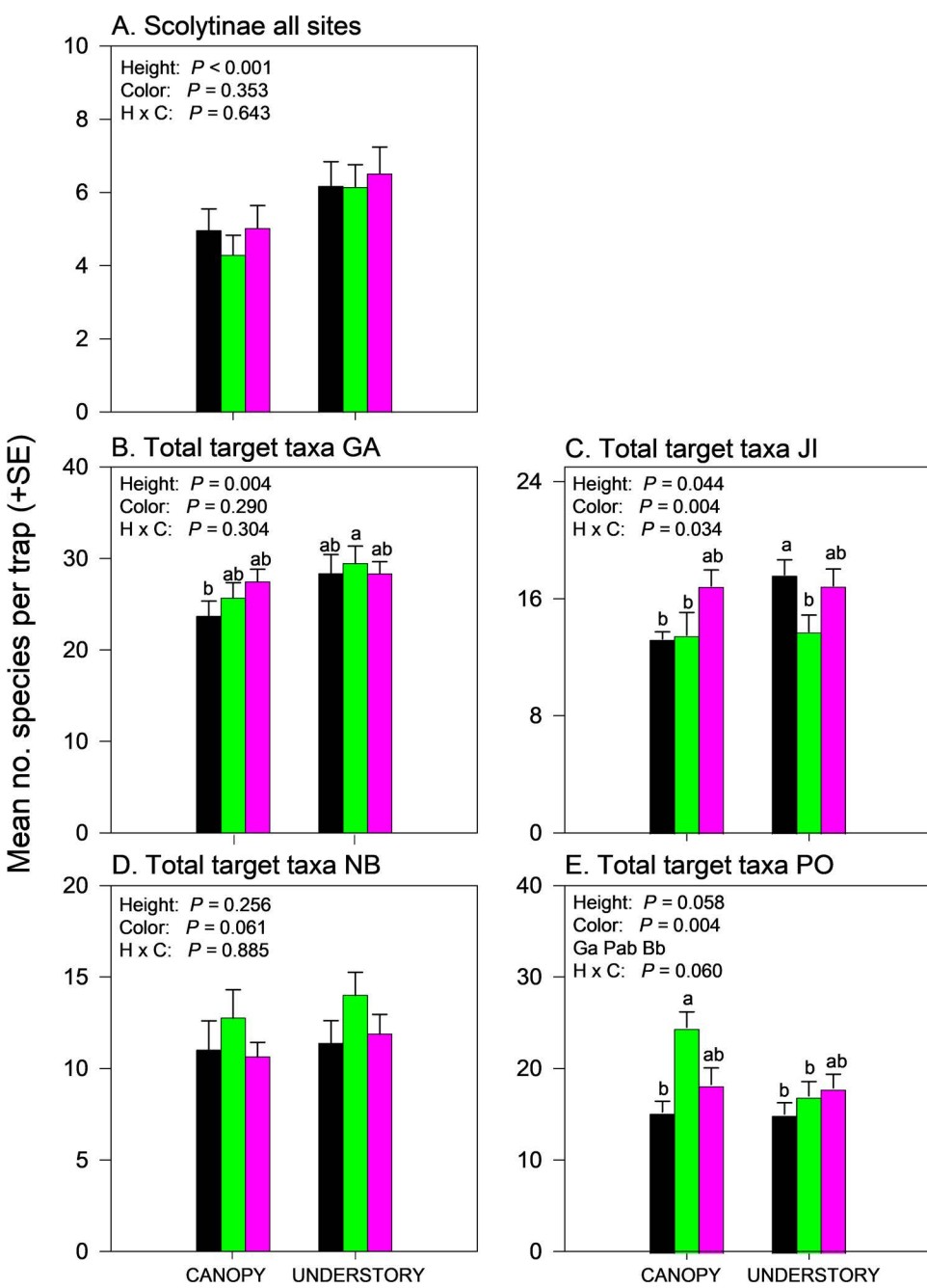

**Fig 8. Effect of trap color (black, green, purple) and trap height (canopy, understory) and their interaction on mean ( ±SE) number of species of bark and woodboring beetles captured in multiple funnel traps: A) Scolytinae, all sites pooled; B) Total target taxa, Georgia, USA; C) Total target taxa, Jilin, China; D) Total target taxa, New Brunswick, Canada; and E) Total target taxa, Białowiez˙a, Poland. Different lowercase letters associated with means and color abbreviations (i.e., B, G, P) indicate significant differences (Tukey-Kramer test on least square means, _P_ ≤ 0.05).**

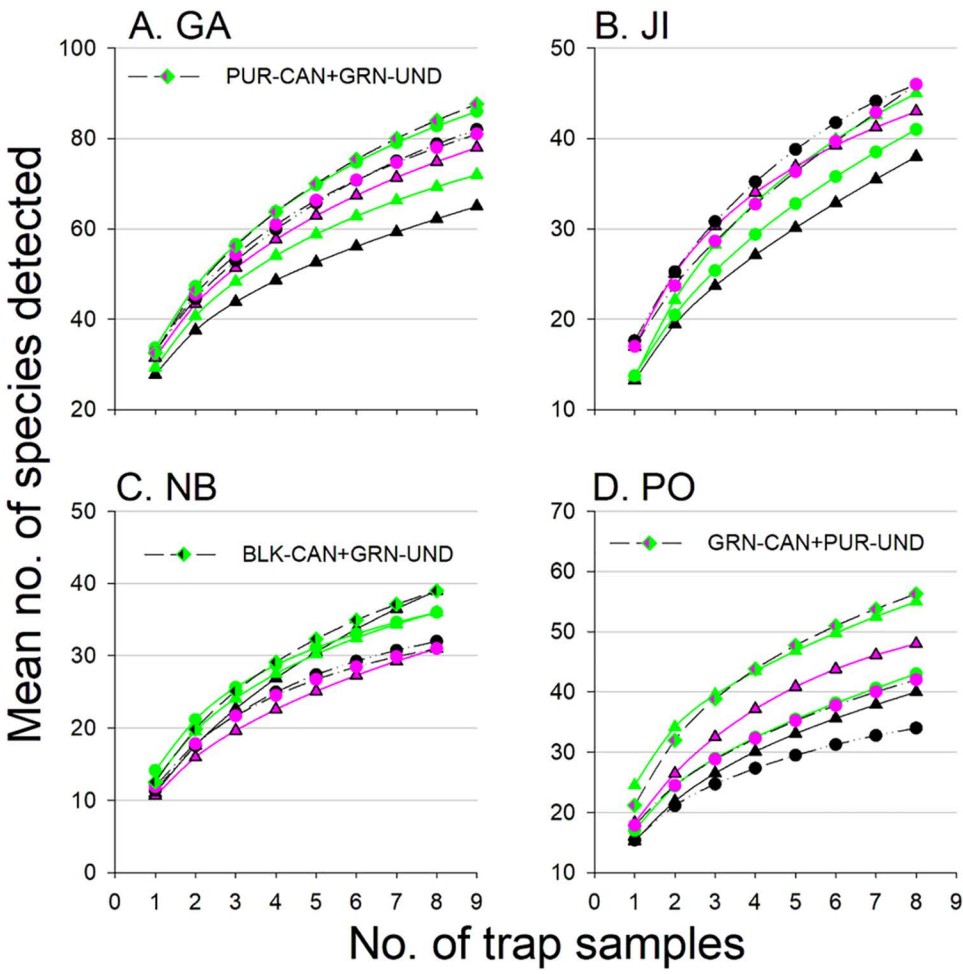

**Fig 9. Species accumulation curves showing the number of species of Buprestidae, Cerambycidae, Disteniidae, and Scolytinae detected per number of traps, in black (BLK), green (GRN), or purple (PUR) 12-funnel traps placed in the forest canopy (triangles) or understory (dots) in: A) Georgia, USA; B) Jilin, China; C) New Brunswick, Canada; and D) Białowieża, Poland.** Also shown for NB, PO, and GA, is the species accumulation curve for the pair of trap height-trap color combinations that detected the most target taxa (in 8 traps in NB and PO, and 9 traps in GA); this curve is not shown for JI because the best pair of treatments detected no more species (45.9) than did the single treatments of black or purple understory traps (46 species in 8 traps).

general there was clear separation of communities between canopy and understory traps, and to a lesser extent by trap color (Fig 10). These patterns were strongest in the data from GA (Fig 10A) and PO (Fig 10D) but also apparent in the data from NB (Fig 10C), and to a lesser extent from JI (Fig 10B). In the upper canopy, black and purple traps caught similar communities of beetles at all sites, but green traps caught different communities than black or purple traps in both PO and GA. There were no substantive differences in communities among trap colors in the lower canopy.

The indicator species analysis showed that 12 species (2 buprestids, 8 cerambycids, 2 scolytines) were more prevalent (i.e., abundant and constant) in canopy traps and 19 species (12 cerambycids, 7 scolytines) were more prevalent in understory traps than would be expected by chance ($P \leq 0.001$) (Table 5). Fewer species were indicators of trap color. Five species were prevalent in green traps (*A. laticornis,* IV = 49.0, *P* = 0.0002; *A. masculinus* Horn, IV = 59.6, *P* = 0.002; *Agrilus olivicolor* Kiesenwetter, IV = 56.2, *P* = 0.0002; *Agrilus sibericus* Obenberger, IV = 43.8, *P* = 0.0004; *Agrilus* sp., IV = 74.5, *P* = 0.0002), only one species was prevalent in black traps (*Neoclytus scutellaris* (Olivier), IV = 42.2, *P* = 0.0010), and none

**Table 3. Species richness of bark beetles and wood boring beetles (BBWB) (Buprestidae, Cerambycidae, Scolytinae) captured in traps of different colors [Black (B), Green (G), Purple (P)] and heights [Canopy (C), Understory (U)] in: Georgia, USA; Jilin, China; New Brunswick, Canada; and Białowieżˋa, Poland in 2015.**

| Trap Color-Height | Georgia | | | Jilin | | | New Brunswick | | | Poland | | |
|---|---|---|---|---|---|---|---|---|---|---|---|---|
| | ND | Ob/Rar | CBE | ND | Ob/Rar | CBE | ND | Ob/Rar | CBE | ND | Ob/Rar | CBE |
| B-C | 42 | 65.0 | 83.9 | 61 | 38.0 | 48.0 | 33 | 39.0* | 51.1* | 44 | 40.0 | 48.9 |
| G-C | 21 | 72.0 | 78.0 | 26 | 45.0 | 45.2 | 20 | 36.0 | 38.7 | 35 | 55.0 | 55.0 |
| P-C | 24 | 78.0 | 87.0 | 22 | 43.0 | 35.8 | 32 | 31.0 | 37.8 | 14 | 48.0 | 46.9 |
| B-U | 20 | 82.0 | 89.1 | 13 | 46.0 | 40.4 | 16 | 32.0 | 33.0 | 16 | 34.0 | 31.6 |
| G-U | 19 | 86.0 | 92.8 | 42 | 41.0 | 44.0 | 20 | 36.0 | 36.8 | 55 | 43.0 | 58.7 |
| P-U | 24 | 81.0 | 89.0 | 75 | 46.0 | 68.6* | 13 | 31.0 | 30.9 | 10 | 42.0 | 42.9 |
| *Avg. 1 color x 1 height* | *25.0* | *77.3* | *86.6* | *39.8* | *43.2* | *47.0* | *22.3* | *34.2* | *38.1* | *29.0* | *43.7* | *47.3* |
| B-C+G-C | 24 | 71.5 | 85.6 | 34 | 41.4 | 43.0 | 10 | 37.5 | 46.5 | 32 | 51.2 | 53.0 |
| B-C+P-C | 34 | 71.8 | 94.9 | 25 | 40.9 | 36.7 | 19 | 34.5 | 45.4 | 18 | 45.5 | 51.5 |
| G-C+P-C | 11 | 75.4 | 82.1 | 23 | 45.6 | 42.9 | 19 | 34.1 | 41.1 | 30 | 56.3* | 60.4* |
| B-U+G-U | 19 | 86.1 | 105* | 24 | 45.9 | 46.0 | 31 | 35.8 | 38.2 | 14 | 38.8 | 39.1 |
| B-U+P-U | 13 | 81.3 | 91.7 | 23 | 45.3 | 40.9 | 19 | 33.2 | 40.2 | 31 | 37.7 | 36.9 |
| G-U+P-U | 16 | 85.4 | 99.4 | 29 | 44.6 | 46.6 | 20 | 36.7 | 43.4 | 14 | 43.2 | 45.4 |
| *Avg. 2 colors x 1 height* | *19.5* | *79.4* | *93.1* | *26.3* | *44.0* | *42.7* | *19.7* | *35.3* | *42.5* | *23.2* | *45.5* | *47.7* |
| B-C+B-U | 14 | 77.2 | 89.4 | 38 | 43.1 | 41.2 | 15 | 36.9 | 46.3 | 25 | 40.1 | 44.1 |
| G-C+G-U | 17 | 84.1 | 99.5 | 32 | 44.5 | 47.8 | 18 | 36.9 | 40.5 | 25 | 55.8 | 57.6 |
| P-C+P-U | 21 | 82.3 | 95.4 | 16 | 44.4 | 38.8 | 21 | 32.7 | 41.3 | 12 | 52.3 | 59.5 |
| *Avg. 1 color x 2 heights* | *17.3* | *80.2* | *94.8* | *28.7* | *44.0* | *42.6* | *18.0* | *35.5* | *42.7* | *20.7* | *49.4* | *53.7* |
| B-C+G-U | 24 | 81.4 | 103 | 49 | 40.1 | 43.3 | 21 | 39.0* | 47.7 | 24 | 45.3 | 52.5 |
| B-C+P-U | 19 | 76.9 | 90.5 | 42 | 42.1 | 43.7 | 26 | 36.1 | 49.1 | 15 | 44.9 | 50.5 |
| G-C+B-U | 19 | 84.0 | 105 | 19 | 46.7* | 44.4 | 8 | 37.3 | 41.6 | 15 | 51.9 | 52.0 |
| G-C+P-U | 19 | 83.4 | 101 | 32 | 46.4 | 48.4 | 18 | 36.6 | 41.8 | 30 | 56.3* | 60.4* |
| P-C+B-U | 17 | 84.0 | 102 | 35 | 44.8 | 38.4 | 31 | 33.9 | 46.3 | 11 | 47.4 | 52.1 |
| P-C+G-U | 15 | 87.6* | 101 | 20 | 45.1 | 43.5 | 9 | 36.6 | 40.2 | 15 | 52.1 | 58.0 |
| *Avg. 2 colors x 2 heights* | *18.8* | *82.9* | *100* | *32.8* | *44.2* | *43.6* | *18.8* | *36.6* | *44.5* | *18.3* | *49.7* | *54.3* |

*Within each site, asterisks denote the treatment or treatment pair with the highest value of observed/estimated species richness. ND = the estimated minimum percentage of BBWB species present at a site that were not detected in a sample of 9 traps (Georgia) or 8 traps (other sites) [ND = (1 – sample completeness at order q = 0)*100]; Ob = actual species richness observed in sample of 9 traps (Georgia) or 8 traps (other sites) for single trap color-trap height treatments; Rar = species richness detected by different combinations of two color-height treatments, interpolated by rarefaction for samples of 9 traps (Georgia) or 8 traps (other sites); CBE = coverage-based estimate of true species richness for a standardized fraction of individuals. Values were determined using the iNEXT 4 steps program and incidence-based data. Trap color-height combinations are ordered top to bottom in categories of least diverse (i.e., one trap color at one trap height) to most diverse (i.e., two trap colors, one in the canopy and one in the understory).

were prevalent in purple traps. When ISA was run with trap height-trap color combination as the grouping variable, 10 species were prevalent in green canopy traps (7 buprestids in the subfamily Agrilinae, 3 cerambycids), two species were prevalent in purple canopy traps (*A. sulcicollis,* Agrilinae; and *Hylocurus rudis* (LeConte), Scolytinae), and one species was prevalent in black canopy traps (*P. detritus,* Cerambycinae) ([Table 6]). Four species were prevalent in black understory traps (all cerambycids), one species was prevalent in green understory traps (*Alosterna tabacicolor* (DeGeer), Lepturinae), and one species was prevalent in purple understory traps (*T. fuscum,* Spondylidinae) ([Table 6]).

## Discussion

Our results demonstrate that trap height, trap color, and their interaction can significantly affect the species richness, species community, and rate of detecting BBWB (i.e., Buprestidae, Cerambycidae, Curculionidae: Scolytinae) in

**Table 4. Results of permutational multivariate analysis of variation for the effect of trap color and trap height on position of bark and wood-boring beetle communities in multivariate space.**

| Location | term | df | Sum of squares | Mean square | F* | R² | p |
|---|---|---|---|---|---|---|---|
| New Brunswick, Canada | height | 1 | 0.508 | 0.5080 | 2.740 | 0.0549 | **0.005** |
| | color | 2 | 0.621 | 0.3100 | 1.680 | 0.0671 | **0.005** |
| | height x color | 2 | 0.345 | 0.1720 | 0.930 | 0.0372 | 0.400 |
| | Residuals | 42 | 7.780 | 0.1850 | | 0.8410 | |
| | Total | 47 | 9.250 | | | 1.0000 | |
| Jilin, China | height | 1 | 0.258 | 0.2580 | 3.530 | 0.0554 | **0.010** |
| | color | 2 | 1.190 | 0.5950 | 8.140 | 0.2560 | **0.005** |
| | height x color | 2 | 0.132 | 0.0661 | 0.904 | 0.0284 | 0.500 |
| | Residuals | 42 | 3.070 | 0.0731 | | 0.6600 | |
| | Total | 47 | 4.650 | | | 1.0000 | |
| Białowieża, Poland | height | 1 | 4.460 | 4.4600 | 36.000 | 0.3780 | **0.005** |
| | color | 2 | 1.230 | 0.6150 | 4.960 | 0.1040 | **0.005** |
| | height x color | 2 | 0.911 | 0.4560 | 3.670 | 0.0772 | **0.010** |
| | Residuals | 42 | 5.210 | 0.1240 | | 0.4410 | |
| | Total | 47 | 11.800 | | | 1.0000 | |
| Georgia, United States | height | 1 | 1.760 | 1.7600 | 30.100 | 0.3330 | **0.005** |
| | color | 2 | 0.505 | 0.2520 | 4.310 | 0.0953 | **0.005** |
| | height x color | 2 | 0.216 | 0.1080 | 1.850 | 0.0408 | 0.095 |
| | Residuals | 48 | 2.810 | 0.0586 | | 0.5310 | |
| | Total | 53 | 5.300 | | | 1.0000 | |

*F and p values determined from 999 permutations [86] of the model. R² and significance of model terms determined by sequential addition

pheromone-baited multiple funnel traps. Thus, both factors should be considered when designing surveillance and monitoring programs for non-native BBWB. Replicating the same experimental design in North America, Europe and Asia revealed that the influence of trap colour and trap height were relatively consistent among continents for some taxa (e.g., Buprestidae: Agrilinae, Chrysochroinae; and Curculionidae: Scolytinae) but not for other taxa (Cerambycidae: Cerambycinae, Lamiinae, Lepturinae, Prioninae, Spondylidinae). This likely reflects variation in foraging habits and preferences of the BBWB species assemblages present at each site [31,33] as well as differences among sites in forest structure and tree species composition that can influence the effects of trap placement or color on BBWB trap catches [29,104–106].

We observed significant interactions between trap height and trap color on species richness of jewel beetles in the subfamilies Agrilinae and Chrysochroinae, confirming results of an earlier study [25]. Most Agrilinae species were captured in green canopy traps (Fig 2E) whereas most Chrysochroinae species were captured in purple canopy traps (Fig 2F). Indicator species analysis supported the trend for greater capture of Agrilinae species in green canopy traps, with seven species significantly more prevalent in green canopy traps and only one species (*A. sulcicollis*) significantly more prevalent in purple canopy traps (Table 6). Black traps performed relatively poorly at detecting jewel beetles. Greater efficacy of canopy *vs.* understory traps for detecting Agrilinae species has been previously observed [29,31,37,53,107–109] and may be due in part to a preference of many *Agrilus* species for sun-exposed microhabitats [106,110]. *Agrilus planipennis* display greater flight activity, landing rate, and foliar feeding in the upper crown of host trees compared to the lower crown [111–113], and use visual stimuli to locate females basking on the upper surface of leaves in sun-exposed areas of the crown [111]. Similar visually-mediated mate-seeking behavior has been observed in male *Agrilus subcinctus* Gory and *A. cyanescens* (Ratzeburg) [114], as well as *A. angustulus* (Illiger), *A. biguttatus* (Fabr.), and *A. sulcicollis* [115].

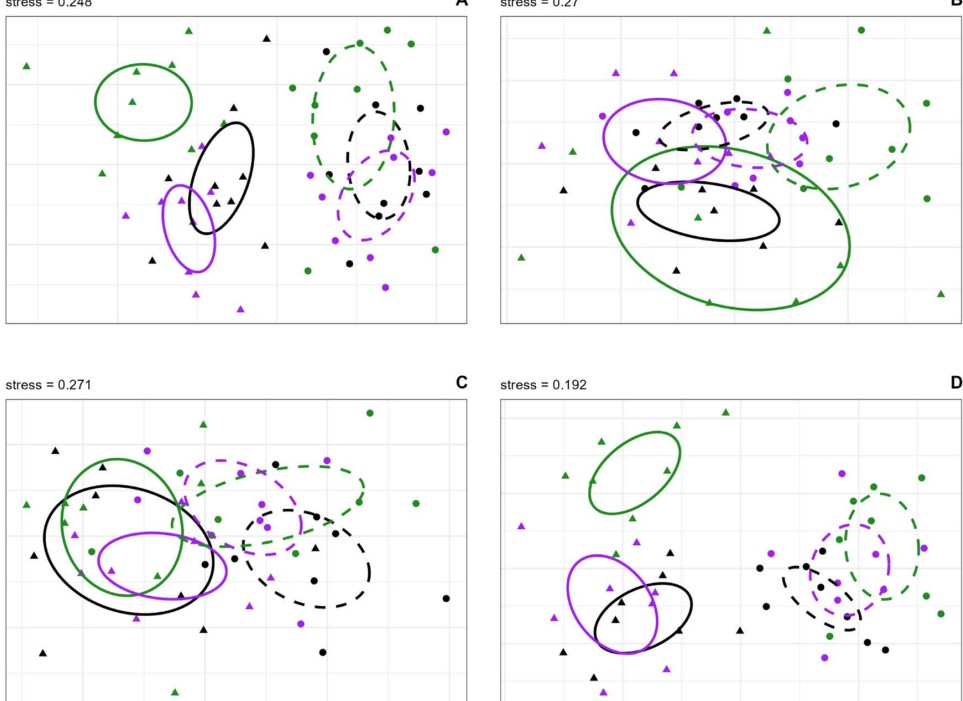

**Fig 10. NMDS ordination showing the community structure of bark beetles and woodboring beetles collected in black, green and purple multiple funnel traps hung in the upper canopy (upward pointing triangles, solid lines) and the lower canopy (downward pointing triangles, broken lines) in: A) Georgia, USA; B) Jilin, China; C) New Brunswick, Canada; and D) Białowiezˇa, Poland, showing 95% confidence interval ellipses. The amount of overlap in the confidence ellipse indicates the degree of similarity among communities caught in the different trap types.**

A preference of jewel beetles for green or purple traps has been observed in other studies [37,51]. Extensive research on the development of tools for survey and monitoring of the invasive emerald ash borer showed that *A. planipennis* eyes were sensitive to ultraviolet, red, blue, and green wavelengths and that green and purple traps caught more individuals than other tested colors [38,45,46,53,63,116]. Subsequent studies showed that green or purple traps were also effective at detecting other species of jewel beetles [25,37,47–52]. Attraction to purple traps has been previously observed for *Dicerca* spp. [51], as well as *A. planipennis* [46], *Agrilus viridis* (L.) [49], *A. biguttatus* [54], *A. auroguttatus* Schaeffer [117], *A. sulcicollis, A. bilineatus* (Weber) [118], *Buprestis lineata* Fabricius [56], and *Chrysobothris* spp. (Buprestinae) [37,44,50,119–121]. A recent study using artificial neural networks (ANNs) suggests that jewel beetles may discriminate leaves (food) and bark (oviposition sites) based on opposing inputs from green-, red-, and blue-sensitive photoreceptors: as signals from green-sensitive photoreceptors increase, stimuli are classified as leaves, and as signals from red- and blue-sensitive photoreceptors increase, stimuli are classified as bark [122]. The same study showed that the ANNs classified green traps as leaves and purple traps as bark [122].

We captured only two species and four specimens of *Anthaxia* spp. suggesting they were either rare at our sites or that none of our trap color-height combinations were very effective at detecting them. The latter is more likely the case because Cavaletto et al. found that *Anthaxia* spp. were more attracted to yellow traps than green traps [37]. It is also possible that the semiochemical lures on the traps deterred capture of *Anthaxia* spp. Santoiemma et al. [87] found that mean catch of *Anthaxia constricticollis* Bílý was significantly greater in unbaited green multiple-funnel traps than in the same traps baited with UHR ethanol plus either *E,Z*-fuscumol and *E,Z*-fuscumol acetate or racemic 3-hydroxyhexan-2-one, racemic 3-hydroxyoctan-2-one, and racemic *syn*-2,3-hexanediol.

**Table 5. Results of blocked indicator species analysis of bark and woodboring beetles captured in black, green or purple multiple funnel traps placed in the canopy and understory of mixed hardwood-coniferous forests in Georgia, U.S.A. (GA), Jilin, China (JI), New Brunswick, Canada (NB, or Białowiez̒a, Poland (PO). The grouping variable was trap height. The larger the IV value, the more prevalent (abundant and constant) the species is within its associated trap height.**

| Trap height | Family | Subfamily | Species | Site | IV | P* |
|---|---|---|---|---|---|---|
| **Canopy** | Buprestidae | Agrilinae | *Agrilus sulcicollis* Lacordaire | PO | 62.1 | 0.0002 |
| | | Chrysochroinae | *Dicerca asperata* Laporte & Gory | GA | 44.4 | 0.0004 |
| | Cerambycidae | Cerambycinae | *Clytus tropicus* (Panzer) | PO | 76.7 | 0.0002 |
| | | | *Neoclytus jouteli* Davis | GA | 80.0 | 0.0002 |
| | | | *Paralaphidion aspersum* (Haldeman) | GA | 58.8 | 0.0002 |
| | | | *Plagionotus detritus* (L.) | PO | 79.2 | 0.0002 |
| | | Lamiinae | *Styloleptus biustus* (LeConte) | GA | 68.5 | 0.0004 |
| | | | *Urgleptes signatus* (LeConte) | NB | 50.0 | 0.0010 |
| | | Lepturinae | *Grammoptera abdominalis* (Stephens) | PO | 45.4 | 0.0008 |
| | | Prioninae | *Sphenostethus taslei* (Buquet) | GA | 51.7 | 0.0002 |
| | Curculionidae | Scolytinae | *Cnestus mutilatus* (Blandford) | GA | 85.5 | 0.0002 |
| | | | *Hylocuris rudis* (LeConte) | GA | 76.2 | 0.0002 |
| **Understory** | Cerambycidae | Cerambycinae | *Cyrtoclytus capra* (Germar) | JI | 74.3 | 0.0004 |
| | | | *Elaphidion mucronatum* (Say) | GA | 81.4 | 0.0002 |
| | | | *Heterachtes quadrimaculatus* Haldeman | GA | 52.4 | 0.0006 |
| | | | *Molorchus bimaculatus* Say | NB | 44.6 | 0.0002 |
| | | | *Neoclytus acuminatus* (Fabricius) | GA | 85.3 | 0.0002 |
| | | | *Neoclytus mucronatus* (Fabricius) | GA | 72.9 | 0.0002 |
| | | | *Xylotrechus colonus* (Fabricius) | GA | 88.2 | 0.0002 |
| | | Lamiinae | *Graphisurus fasciatus* (DeGeer) | GA | 70.7 | 0.0002 |
| | | Lepturinae | *Bellamira scalaris* (Say) | GA | 49.6 | 0.0006 |
| | | Prioninae | *Prionus insularis* Motschulsky | JI | 65.9 | 0.0002 |
| | | Spondylidinae | *Tetropium castaneum* (L.) | PO | 61.2 | 0.0002 |
| | | | *Tetropium fuscum* (Fabricius) | PO | 90.4 | 0.0002 |
| | Curculionidae | Scolytinae | *Ambrosiophilus atratus* Eichhoff | GA | 66.7 | 0.0002 |
| | | | *Anisandrus dispar* (Fabricius) | PO | 96.1 | 0.0002 |
| | | | *Dryoxylon onoharaense* (Murayama) | GA | 76.6 | 0.0002 |
| | | | *Scolytus rugulosus* (Müller) | NB | 71.0 | 0.0002 |
| | | | *Xyleborinus saxesenii* (Ratzeburg) | GA | 85.8 | 0.0002 |
| | | | *Xyleborus bispinatus* Eichhofff | GA | 37.0 | 0.0002 |
| | | | *Xylosandrus crassiusculus* (Motschulsky) | GA | 89.5 | 0.0002 |

*p values determined from Monte Carlo randomization test with 4999 permutations, calculating for each species, the proportion of permutations in which the randomized maximum IV was greater than or equal to the observed maximum IV. *P*-values of 0.001 or smaller for a species and grouping indicate that species is more abundant and constant in that grouping than would be expected by chance.

Our results agree with other studies that found green canopy traps effective for detecting *Agrilus* [25,45,123] and suggest that of the six trap color-height combinations we tested, green canopy traps would be the most suitable for surveillance of non-native *Agrilus* spp. Seven species of jewel beetles captured in Jilin represent the earliest records from that province [124,125]. Five of these new species records (*Agrilus euonymi* Toyama, *A. peregrinus* Kiesenwetter, *A. sibericus* Obenberger, *A. smaragdinus* Lacordaire, and *A. viduus* Kerremans) were reported separately [126] and *Coraebus aequalipennis* Fairmaire is reported from Jilin province here for the first time; all but 5 of the 37 specimens of these six Agrilinae species were captured in green traps (Table S3). The seventh species, *Lamprodila virgata* Motschulsky (Chrysochroinae),

**Table 6. Results of blocked indicator species analysis of bark and woodboring beetles captured in black, green or purple multiple funnel traps placed in the canopy and understory of mixed hardwood-coniferous forests in Georgia, U.S.A. (GA), Jilin, China (JI), New Brunswick, Canada (NB, or Białowiez˙a, Poland (PO). The grouping variable was trap height-trap color combination. The larger the IV value, the more prevalent (abundant and constant) the species is within the associated trap height-trap color combination.**

| Trap height | Trap color | Family | Subfamily | Species | Site | IV | *P*\* |
|---|---|---|---|---|---|---|---|
| **Canopy** | **Black** | Cerambycidae | Cerambycinae | *Plagionotus detritus* (L.) | PO | 36.0 | 0.0004 |
| | **Green** | Buprestidae | Agrilinae | *Agrilus angustulus* (Illiger) | PO | 64.8 | 0.0002 |
| | | | | *Agrilus bilineatus* (Fabricius) | GA | 65.4 | 0.0002 |
| | | | | *Agrilus laticornis* (Illiger) | PO | 98.1 | 0.0002 |
| | | | | *Agrilus obscuricollis* Keisenwetter | PO | 64.3 | 0.0006 |
| | | | | *Agrilus* sp. | GA | 52.3 | 0.0004 |
| | | | | *Brachys ovatus* (Weber) | GA | 54.7 | 0.0004 |
| | | | | *Trachys minutus* (L.) | PO | 75.0 | 0.0002 |
| | | Cerambycidae | Cerambycinae | *Molorchus minor* (L.) | PO | 56.2 | 0.0008 |
| | | | | *Neoclytus jouteli* Davis | GA | 57.2 | 0.0002 |
| | | | Lepturinae | *Grammoptera ustulata* (Schaller) | PO | 73.3 | 0.0002 |
| | **Purple** | Buprestidae | Agrilinae | *Agrilus sulcicollis* Lacordaire | PO | 56.9 | 0.0004 |
| | | Curculionidae | Scolytinae | *Hylocurus rudis* (LeConte) | GA | 35.6 | 0.0010 |
| **Understory** | **Black** | Cerambycidae | Cerambycinae | *Neoclytus acuminatus* (Fabricius) | GA | 34.7 | 0.0002 |
| | | | | *Neoclytus mucronatus* (Fabricius) | GA | 31.0 | 0.0002 |
| | | | Lamiinae | *Aegomorphus quadrigibbus* (Say) | GA | 41.3 | 0.0004 |
| | | | Lepturinae | *Bellamira scalaris* (Say) | GA | 60.4 | 0.0004 |
| | **Green** | | Lepturinae | *Alosterna tabacicolor* (DeGeer) | PO | 72.9 | 0.0004 |
| | **Purple** | | Spondylidinae | *Tetropium fuscum* (Fabricius) | PO | 63.2 | 0.0002 |

\**P* values determined from Monte Carlo randomization test with 4999 permutations, calculating for each species, the proportion of permutations in which the randomized maximum IV was greater than or equal to the observed maximum IV. *P*-values of 0.001 or smaller for a species and grouping indicate that species is more abundant and constant in that grouping than would be expected by chance.

was captured again in Jilin in 2021 and recently reported by Santoiemma et al. [87]. Further evidence of the suitability of green, Fluon-treated multiple funnel canopy traps for detecting *Agrilus* spp. is that they provided the first distributional records of *Agrilus hastulifer* (Ratzeburg) in Poland [127], *A. graminis* Kiesenwetter in the Białowieża Primeval Forest, Poland [108], *A. juglandis* Knull, *A. osborni* Knull, and *A. masculinus* in New Brunswick, Canada [128], *A. arcuatus* (Say) and *A. obsoletoguttatus* Gory in Nova Scotia, Canada [129], and *A. masculinus* on Prince Edward Island, Canada [130]. A recent study found that green sticky prism canopy traps captured more species and more specimens of *Agrilus* spp. than did Fluon-treated green multiple funnel canopy traps [123], indicating they would be more efficacious for *Agrilus* surveillance. However, funnel traps have some practical advantages to sticky traps; they can be re-used for several years, take less time to check and collect insects from in the field, and are less susceptible to reductions in trapping efficiency due to accumulation of dust, pollen and other debris on trap surfaces [63,123].

Our results clearly demonstrate the benefits of placing traps in both the canopy and understory for detection of Scolytinae. Species richness of Scolytinae was greater in understory traps than in canopy traps, as reported in many [24,30–32,104,131–133] but not all [26,34,104] studies. Indicator species analysis showed that seven scolytine species were significantly more prevalent in understory traps and two species were significantly more prevalent in canopy traps (Table 5). The effect of trap height on species richness of Scolytinae is also known to vary among feeding guilds; species richness of ambrosia beetles decreases with increasing trap height whereas that of phloem-feeding bark beetles increases [31,105,134,135]. Of the 13 scolytine species whose rate of detection was not independent of trap height-color combination in our study, nine species of ambrosia beetles and one species of bark beetle were detected more frequently

in the understory than the canopy (Table S3). However, the reverse was true for two species of bark beetle [*Pityogenes chalcographus* (L.), *Scolytus carpini* (Ratzeburg)], and one species of ambrosia beetle [*Xyleborus monographus* (Fabricius)] (Table S3). Native to Europe and parts of Africa and Asia, *X. monographus* is established in California where it infests *Quercus* and other Fagaceae, attacking large diameter trunks of downed trees and branches as small as 6.4 cm in diameter in the upper crowns of apparently healthy oak trees [136]. Mean catch of *Cnestus mutilatus* (Blandford), an invasive ambrosia beetle species in the United States [137] as well as parts of Europe [138], was greater in the canopy than the understory, as previously observed [28,31].

We found no effect of trap color on species richness of Scolytinae captured, and no effect on abundance of individual species of Scolytinae except for *X. crassiusculus* for which mean catch was greater in black traps than green traps. This is contrary to most evidence in the literature that suggests dark colored traps, like black or purple, are better for survey of bark and ambrosia beetles. In a previous study, Scolytinae species richness was significantly greater in purple traps than green traps in the understory but not in the canopy, and only in reforested areas; there was no effect of trap color on Scolytinae catches in semi-natural forests [52]. Miller [56] compared catches of saproxylic beetles in modified 10-unit multiple funnel traps of the same black, green, and purple colors used in the present study, set in the forest understory, and observed greater catches in black or purple traps than in green traps for eight species of Scolytinae, including *X. crassiusculus.* The same study found that purple traps captured more *C. mutilatus* than black traps did [56], whereas we observed no effect of trap color on catch of *C. mutilatus.* A broader range of trap colors (i.e., black, green, purple, yellow, red, blue, gray, brown) also had no effect on Scolytinae species richness but significantly more specimens of some ambrosia beetles, e.g., *Xyleborinus saxesenii* (Ratzeburg) and *X. monographus*, were caught in black traps than in green or yellow traps [37]. Other studies found no differences in catch of ambrosia beetles [139,140] or bark beetles [139,141] among colors like black, green, red, brown or gray, but found significantly lower catches in white *vs.* black traps [139,141] especially when traps were baited with pheromones and host volatiles [35,142]. Catch of the bark beetle, *Dendroctonus frontalis* Zimmermann, was greatest in traps that presented a dark silhouette with low reflectance regardless of color [141].

Our results clearly support the importance of placing traps in both the canopy and understory for survey of longhorn beetles, in agreement with many other studies [24,25,27–29,33,34,135,143,144]. In our study, however, the effect of trap height on species richness of Cerambycidae captured varied among sites, subfamilies, and species. There were no effects of trap height on cerambycid species richness captured in NB and JI, but trap color and trap height interacted to affect the number of species captured in GA and PO. For example, green canopy traps captured more cerambycid species than any other treatment in PO but significantly fewer species than green understory traps in GA. Variation among sites in vertical stratification of longhorn beetles is common, with reports of greater [24,25,29,34,104,131,135], similar [24,27,33,104,132,143,144], or lower [26,32] species richness in the canopy compared to the understory. The indicator species analysis showed that nine cerambycid species were significantly more prevalent in canopy traps and twelve cerambycid species were significantly more prevalent in understory traps (Table 5). This suggests the species richness of cerambycids in different parts of the canopy is likely influenced by the species-specific foraging habits of the cerambycid species present at the survey site, as well as site-specific factors such as forest type, stand structure, and edge effects [20] which is then reflected in the species assemblages captured by the traps. Twig girdlers and other species that breed in small branches spend more time foraging in the canopy whereas those that breed in logs and stumps are more active in the understory [24,131,144]. The trends we observed agreed with those reported in the literature in many cases, e.g., greatest catches of *P. testaceus* [24] and *P. detritus* [24,25] in the canopy and greatest catches of *N. acuminatus* [28,33] and *T. fuscum* in the understory [24]. However, we observed greatest catches of *Neoclytus mucronatus* (Fabricius) in the understory whereas others found the opposite [143] or no trend [28].

Our results show that efficacy of multiple funnel traps for survey of longhorn beetles is affected by trap color and that color preferences vary among species. This indicates that deploying more than one trap color should increase the diversity of species captured. The effect of trap color on cerambycid species richness captured also varied among sites, with

black or purple traps detecting more species than green traps in JI (Fig 3B), and green canopy traps detecting more species than any other treatment except purple canopy traps in PO (Fig 3D). Trends within subfamilies were also not consistent among sites, e.g., purple traps caught more species of Lepturinae than green traps in JI (Fig 6B), but the reverse was true in NB (Fig 6C) and PO (Fig 6D). Previous studies that compared cerambycid catches in green and purple multiple funnel traps [25], or black, green and purple multiple funnel traps [51], found no significant effects of trap color on species richness captured. However, in a comparison of eight different colors of panel traps, Cavaletto et al. found that yellow and blue traps captured more species of longhorn beetles than did black traps [36]. The same study found that black traps captured significantly fewer species of: 1) Cerambycinae than did yellow traps; 2) Lamiinae than did brown or red traps; and 3) Lepturinae than did yellow, blue or green traps [36]. Effects of trap color on mean catch of individual cerambycid species agreed with previous findings for some species but not others. We observed greater catch of *P. detritus* in purple *vs.* green traps, as did Rassati et al. [25] and greater catches of *N. acuminatus* and *N. mucronatus* in black traps than in green traps, as did Miller [56], whereas Skvarla and Dowling [51] found no differences in longhorn beetle catches among green, black or purple traps. Differences in color response between our study and Skvarla and Dowling's [51] could be due to a number of factors. They used unbaited traps whereas we baited ours with semiochemicals attractive to several longhorn beetle species. Longhorn beetles display greater discrimination among visual cues such as trap color when traps are baited with attractive semiochemicals [142].

As predicted, the communities of BBWB captured in the canopy differed from those in understory (Fig 10), in agreement with many previous studies [24,26–28,31,34,131,144]. The clearer separation of BBWB communities by trap height in PO and GA, compared to JI and NB (Fig 10) may have been partially due to the greater distance between understory and canopy traps in the former sites. Understory traps were at similar heights at all sites, but canopy traps were at heights of 11–27 m in PO and 18–23 m in GA compared to 8–16 m in NB and 4–10 m in JI. The height separation between some canopy and understory traps in JI was also reduced because traps were set up along a transect that ran along a hilltop, with traps either along the ridge top or on either side of it. This sometimes resulted in only a slight difference in elevation between an understory trap on the ridge top and a canopy trap located a few meters below the ridge top. However, other studies found clear differences in species composition of saproxylic beetles between traps placed in the understory (at heights of 0.4–1.2 m) and traps at heights as low as 5–7 m in the canopy [30,145]. The separation between canopy and understory species assemblages was less distinct in NB than in GA or PO and this may have been due to edge effects. At the NB site, traps ran along two linear transects, each within 15–25 m wide x 500 m long strips of forest, and although traps were placed as close to the center of the strips to reduce edge effects, they were within 7–12 m of the forest edge and possibly more susceptible to edge effects compared to the other sites. Vertical stratification of BBWB species assemblages is often less distinct along forest edges than inside the forest [26,29,31,106].

Our results suggest that trapping surveys for detection of non-native BBWB may be using too few traps to effectively survey rare species. In North America, trapping surveys for detection of non-native BBWB generally deploy three to six traps per site [15,21]. Our species accumulation curves showed that an 8–9 trap sample fell far short of reaching an asymptote for any combination of trap color and trap height. Even when data from all 48 or 54 traps per site were used, a minimum of 16–26% of BBWB species present at a site were not detected. Thus, increasing the number of traps deployed per site will increase the BBWB species richness captured and the chances of detecting non-native species that may be present. However, the survey budgets of regulatory agencies that deploy traps are often fixed, and so increasing the number of traps comes with tradeoffs: like reducing the number of sites surveyed per year or purchasing fewer semiochemical lures. If we assume there are practical limits on the number of traps regulatory agencies can deploy at a site, it makes sense to use combinations of trap height and color that detect the most species per number of traps deployed. Our prediction that species richness of BBWB detected would be increased by deploying traps in both the canopy and understory and using more than one trap color was supported by the data, although the most effective combination of trap color(s) and height(s) varied among sites. When there is a cap on the total number of traps per survey site, using more than one

trap color-height combination necessarily reduces the number of traps deployed of each combination, and species accumulation curves clearly show that the fewer traps deployed, the fewer species detected [24,29,51,106]. Thus, compared to having eight black traps in the understory, placing four green traps in the canopy and four black traps in the understory will increase the number of jewel beetles detected but reduce the number of species that tend to be detected in black understory traps.

## Conclusions

Our results demonstrate that trap height and color significantly affect the species richness of BBWB detected as well as mean catch per trap of many BBWB species and strongly suggests that trapping surveys for surveillance of non-native BBWB should place traps in both the canopy and understory and use more than one trap color. However, we also found that the combination of trap color and height that detected the most BBWB species was not consistent among sites, so though diversity in trap color and vertical trap placement is important for surveillance, there is no "one size fits all" prescription for maximizing the species richness of BBWB detected. As suggested by Dodds et al. [20], the combination of green canopy traps and either purple or black understory traps would be a good starting point for BBWB trapping surveys and certainly an improvement in terms of detecting *Agrilus* spp. and other jewel beetles compared to using only black traps placed in the understory. However, alternating different combinations of trap colors and height in different years, and increasing the number of traps deployed per site may increase chances of detecting a broader range of BBWB species.

## Supporting Information

**Table S1. Source, percentage purity, lure type, and release rate of semiochemical lures used in trapping experiments testing effects of trap color and trap height on detection of bark and wood boring beetles.**
(DOCX)

**Table S2. Results of generalized linear mixed models testing for interactions between site and treatment (treatments = six different combinations of trap height and trap color) on species richness of target taxa captured in multiple funnel traps in Georgia USA (GA), Jilin, China (JI), New Brunswick, Canada (NB), and Białowieẓa, Poland (PO).** Spondylidinae were captured in JI and PO only and Prioninae were captured in GA and JI only.
(DOCX)

**Table S3. Total catch of Buprestidae, Cerambycidae, Disteniidae, and Curculionidae: Scolytinae in black (Blk), green (Grn), or purple (Pur) multiple-funnel traps placed in the canopy or understory of mixed hardwood-coniferous forests in Georgia, USA (GA), Jilin province, China (JI), New Brunswick, Canada (NB) or Białowieẓa, Poland (PO) in the summer of 2015.**
(XLSX)

**Table S4. Results of generalized linear models testing for the effects of trap height, trap color and their interaction on species richness of target taxa by family and subfamily captured in traps at four sites: Georgia, USA (GA), Jilin, China (JI), New Brunswick, Canada (NB), and Białowieẓa, Poland (PO). Data were analyzed separately by site whenever the preliminary analyses found a significant interaction between site and trap color-height treatment (see Table S2).**
(DOCX)

**Table S5. Results of generalized linear models testing for the effects of trap height, trap color and their interaction on mean (± SE) season catch per trap of bark and woodboring beetles in black, green, or purple**

multiple-funnel traps placed in the canopy (Can) or understory (Und) of trees in mixed deciduous-coniferous forests in 2015 in Georgia, USA (GA), Jilin, China (JI), New Brunswick, Canada (NB), and Białowieża, Poland (PO). (XLSX)

**Table S6. Sample completeness and estimated species richness of bark and wood boring beetles (Buprestidae, Cerambycidae, Scolytinae) detected in the entire sample of 48–54 pheromone-baited multiple funnel traps deployed in Georgia, USA, Jilin, China, New Brunswick, Canada, and Białowieża, Poland.**
(DOCX)

**Fig S1. Results of iNEXT 4-steps analysis comparing six different trap color-trap height treatments and all fifteen possible binary combinations of the same six treatments for estimating species richness of bark and wood boring beetles in Georgia, USA, 2015.** Legend: Each treatment was replicated 9 times for a total of 54 traps. A) sample completeness: values at $q = 0$ estimate the proportion of total species present at a site that were detected in the sample; B) species richness *vs.* number of trap samples, estimated by rarefaction or extrapolation based on sample size; C) observed and asymptotic species richness estimates of species diversity (when $q = 0$); and D) coverage-based species richness for standardized coverage value of $C_{max} = 0.944$. B = black, G = green, P = purple, C = canopy, U = understory. (TIF)

**Fig S2. Results of iNEXT 4-steps analysis comparing six different trap color-trap height treatments and all fifteen possible binary combinations of the same six treatments for estimating species richness of bark and wood boring beetles in New Brunswick, Canada, 2015.** Legend: Each treatment was replicated 8 times for a total of 48 traps. A) sample completeness: values at $q = 0$ estimate the proportion of total species present at a site that were detected in the sample; B) species richness *vs.* number of trap samples, estimated by rarefaction or extrapolation based on sample size; C) observed and asymptotic species richness estimates of species diversity (when $q = 0$); and D) coverage-based species richness for standardized coverage value of $C_{max} = 0.923$. B = black, G = green, P = purple, C = canopy, U = understory. (TIF)

**Fig S3. Results of iNEXT 4-steps analysis comparing six different trap color-trap height treatments and all fifteen possible binary combinations of the same six treatments for estimating species richness of bark and wood boring beetles in Jilin province, P.R. China, 2015.** Legend: Each treatment was replicated 8 times for a total of 48 traps. A) sample completeness: values at $q = 0$ estimate the proportion of total species present at a site that were detected in the sample; B) species richness *vs.* number of trap samples, estimated by rarefaction or extrapolation based on sample size; C) observed and asymptotic species richness estimates of species diversity (when $q = 0$); and D) coverage-based species richness for standardized coverage value of $C_{max} = 0.85$. B = black, G = green, P = purple, C = canopy, U = understory. (TIF)

**Fig S4. Results of iNEXT 4-steps analysis comparing six different trap color-trap height treatments and all fifteen possible binary combinations of the same six treatments for estimating species richness of bark and wood boring beetles in Białowieża, Poland, 2015.** Legend: Each treatment was replicated 8 times for a total of 48 traps. A) sample completeness: values at $q = 0$ estimate the proportion of total species present at a site that were detected in the sample; B) species richness *vs.* number of trap samples, estimated by rarefaction or extrapolation based on sample size; C) observed and asymptotic species richness estimates of species diversity (when $q = 0$); and D) coverage-based species richness for standardized coverage value of $C_{max} = 0.906$. B = black, G = green, P = purple, C = canopy, U = understory. (TIF)

**Fig S5. Results of iNEXT 4-steps analysis estimating species richness of bark and wood boring beetles from 48–54 pheromone-baited multiple funnel traps deployed in Georgia, USA (GA), Jilin, China (JI), New Brunswick,**

**Canada (NB), and Białowieẑa, Poland (PO).** Legend: Equal numbers of black, green, and purple traps were deployed in the understory and canopy at each site. A) sample completeness: values at q = 0 estimate the proportion of total species present at a site that were detected in the sample; B) species richness *vs.* number of trap samples, estimated by rarefaction or extrapolation based on sample size; C) observed and asymptotic species richness estimates of species diversity (when q = 0); and D) coverage-based species richness for standardized coverage value of $C_{max}$ = 0.99.
(TIF)

## Acknowledgments

We thank Krzysztof Sućko, Chris Crowe, Lisa Leachman, and Li Luyao for technical assistance, Eduard Jendek for species determination of jewel beetles captured in Jilin, and Reginald Webster for species determination of beetles in New Brunswick. We also thank the following individuals and institutions for allowing us to conduct trapping experiments on their property: Gerald Redmond in Keswick Ridge, New Brunswick; the administrators of the State-Owned Forest Protection Center of Forestry Experimental Area of Jilin Province, Jiaohe, Jilin Province, China; the Białowieża Forest District in Poland; and the Clybel Wildlife Management Center in Georgia.

## Author contributions

**Conceptualization:** Jon Sweeney.

**Data curation:** Jon Sweeney.

**Formal analysis:** Jon Sweeney, Chantelle Kostanowicz, Christian J. K. MacQuarrie, Vincent Webster.

**Funding acquisition:** Jon Sweeney, Troy Kimoto, Qingfan Meng, Daniel R. Miller.

**Investigation:** Jon Sweeney, Wentao Gao, Jerzy M. Gutowski, Cory Hughes, Chantelle Kostanowicz, Yan Li, Peter Mayo, Qingfan Meng, Tomasz Mokrzycki, Peter Silk, Vincent Webster, Daniel R. Miller.

**Methodology:** Jon Sweeney, Cory Hughes.

**Project administration:** Jon Sweeney.

**Resources:** Jon Sweeney, Jerzy M. Gutowski, Peter Mayo, Qingfan Meng, Tomasz Mokrzycki, Peter Silk, Daniel R. Miller.

**Supervision:** Jon Sweeney, Jerzy M. Gutowski, Cory Hughes, Qingfan Meng, Daniel R. Miller.

**Validation:** Jon Sweeney.

**Visualization:** Jon Sweeney, Christian J. K. MacQuarrie.

**Writing – original draft:** Jon Sweeney.

**Writing – review & editing:** Jon Sweeney, Jerzy M. Gutowski, Cory Hughes, Troy Kimoto, Chantelle Kostanowicz, Yan Li, Christian J. K. MacQuarrie, Peter Mayo, Qingfan Meng, Tomasz Mokrzycki, Peter Silk, Vincent Webster, Daniel R. Miller.

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
