## [Decision Letter · Decision Letter 0]

21 Feb 2025

PONE-D-25-03887Diversity in trap color and height increases species richness of bark and woodboring beetles detected in multiple funnel trapsPLOS ONE

Dear Dr. Sweeney,

Thank you for submitting your manuscript to PLOS ONE. After careful consideration, we feel that it has merit but does not fully meet PLOS ONE’s publication criteria as it currently stands. Therefore, we invite you to submit a revised version of the manuscript that addresses the points raised during the review process.

We look forward to receiving your revised manuscript.

Kind regards,

Ramzi Mansour

Academic Editor

PLOS ONE

Reviewers' comments:

Reviewer's Responses to Questions

**Comments to the Author**

1. Is the manuscript technically sound, and do the data support the conclusions?

Reviewer #1: Yes

Reviewer #2: Yes

2. Has the statistical analysis been performed appropriately and rigorously? 

Reviewer #1: Yes

Reviewer #2: Yes

3. Have the authors made all data underlying the findings in their manuscript fully available?

Reviewer #1: Yes

Reviewer #2: Yes

4. Is the manuscript presented in an intelligible fashion and written in standard English?

Reviewer #1: Yes

Reviewer #2: Yes

5. Review Comments to the Author

Reviewer #1: 

Manuscript Number: PONE-D-25-03887

Manuscript Title: Diversity in trap color and height increases species richness of bark and woodboring beetles detected in multiple funnel traps

Authors: Sweeney et al.

This manuscript summarizes experiments that investigated the influence of trap colors and trap height on captures of bark beetles and woodborers. Three trap colors and two trap heights using almost identical methodology were tested in four countries and results in a very strong experimental design with widely applicable results. The experiments and discussion of results aligns with the objectives in detection surveys that species richness trumps abundance, and the manuscript ends with results that can help improve detection and monitoring efforts for these species. The manuscript is very well prepared and reads well. I found it a bit long, although I found few places where reductions were warranted. I only have a few comments for the authors to consider:

L78: Consider swapping a different word for “forage”, as to me, that links more strongly to habitat, whereas much of that information is unknown for insects captured in the understory. I think “disperse” makes more sense as the BBWB may move through the understory but actually forage/colonize higher on tree boles after landing.

L108: Mentioning “abundance at the family level” seems unnecessary to mention here and could be removed.

L142: I was a bit alarmed when I saw how narrow the forest strips were in NB. It seems like most of the strip would be influenced by edges, especially given how abrupt they were (i.e., adjacent to farm fields). The authors address this briefly in the discussion and that’s probably enough to alleviate concerns. In several ways, it may not matter as even if you were in mostly “edge” habitat, the variables tested were still consistent.

L170: It’s worth noting minimum distance from trap to tree.

Table 3: Looks like the columns were cut. I really like the inclusion of %ND and think it’s a really useful way to look at treatment variation.

L536: Is this “per sample”, or over the course of the trapping season? If the latter works, I think it makes it more meaningful.

L627: It may be worth noting here, or elsewhere, that there’s an assumption that attraction by these species is only to visual cues, no semiochemicals.

Paragraph starting at L631: Here is one place where I think the text could be shortened – eye sensitivity information can just be referenced instead of provided and referenced.

L647: Or they displayed no attraction to selected semiochemicals. Alternatively, you could finish the sentence off by no known response to these semiochemicals.

Reviewer #2: 

The manuscript is very detailed and well-written, and overall, I had only minor comments (minor revision). Several appropriately chosen statistical methods were used to evaluate the results. I believe the authors have provided all relevant data in the manuscript, allowing the reviewer to gain a comprehensive understanding of the study. The English language used in the article is of a high scientific standard.

6. PLOS authors have the option to publish the peer review history of their article (what does this mean? ). If published, this will include your full peer review and any attached files.

**Do you want your identity to be public for this peer review?** For information about this choice, including consent withdrawal, please see our Privacy Policy .

Reviewer #1: No

Reviewer #2: No

---

## [Author Response · Author response to Decision Letter 1]

17 Mar 2025

Response to Reviewer’s comments (our response is in bold font). Please note: The line numbers for new text that we provide in our responses refer to those in the revised manuscript version that includes track changes and all markup.

Reviewer #1:

Manuscript Number: PONE-D-25-03887

Manuscript Title: Diversity in trap color and height increases species richness of bark and woodboring beetles detected in multiple funnel traps

Authors: Sweeney et al.

This manuscript summarizes experiments that investigated the influence of trap colors and trap height on captures of bark beetles and woodborers. Three trap colors and two trap heights using almost identical methodology were tested in four countries and results in a very strong experimental design with widely applicable results. The experiments and discussion of results aligns with the objectives in detection surveys that species richness trumps abundance, and the manuscript ends with results that can help improve detection and monitoring efforts for these species. The manuscript is very well prepared and reads well. I found it a bit long, although I found few places where reductions were warranted. I only have a few comments for the authors to consider:

L78: Consider swapping a different word for “forage”, as to me, that links more strongly to habitat, whereas much of that information is unknown for insects captured in the understory. I think “disperse” makes more sense as the BBWB may move through the understory but actually forage/colonize higher on tree boles after landing.

Agreed. We have changed “forage” to “actively fly” (line 78).

L108: Mentioning “abundance at the family level” seems unnecessary to mention here and could be removed.

We have altered the sentence as follows (lines 106–109) “We also used mean catch per trap of individual species, or their detection rate (proportion of traps that captured at least one specimen), as response variables. rather than abundance at the family level because the latter is largely a product of species richness and abundance of individual species within a given family.”

L142: I was a bit alarmed when I saw how narrow the forest strips were in NB. It seems like most of the strip would be influenced by edges, especially given how abrupt they were (i.e., adjacent to farm fields). The authors address this briefly in the discussion and that’s probably enough to alleviate concerns. In several ways, it may not matter as even if you were in mostly “edge” habitat, the variables tested were still consistent.

No response is deemed necessary.

L170: It’s worth noting minimum distance from trap to tree.

Agreed. We have added the following to the end of the sentence (lines 187–188)

“…and at least 1 m of horizontal distance between the trap and the adjacent trees.”

Table 3: Looks like the columns were cut. I really like the inclusion of %ND and think it’s a really useful way to look at treatment variation.

We apologize for the truncated columns in Table 3. We have edited Table 3 to ensure that no information is cut off by the margins.

L536: Is this “per sample”, or over the course of the trapping season? If the latter works, I think it makes it more meaningful.

We have clarified this sentence. It now reads (lines 567–571): “Depending on the site, using one color of trap in the understory and a different colored trap in the canopy (i.e., 2 colors x 2 heights) detected 1–7 more (observed/rarefied) species over the course of the trapping season than did a single trap color placed in either the understory or canopy (Table 3).”

L627: It may be worth noting here, or elsewhere, that there’s an assumption that attraction by these species is only to visual cues, no semiochemicals.

Agreed. We have added the following sentence to the Methods where we first mention the semiochemical lures used in our study (lines 210–215):

“We assumed these lures would have negligible effects on catches of jewel beetles based on previous studies that found no attraction to ethanol (85,86) or to the longhorned beetle pheromones used in our study (24,25). More recently, Santoiemma et al. (87) found negative effects on mean catches of 6 of 27 species of jewel beetles when UHR ethanol or combinations of ethanol plus cerambycid pheromones were added to unbaited traps; however, the lures had no effect on species richness of jewel beetles captured.”

Paragraph starting at L631: Here is one place where I think the text could be shortened – eye sensitivity information can just be referenced instead of provided and referenced.

We shortened the sentence as follows (lines 665–668): “Extensive research on the development of tools for survey and monitoring of the invasive emerald ash borer showed that A. planipennis eyes were sensitive to ultraviolet, red, blue, and green wavelengths and that green (530–536 nm, 57%) and purple (421 nm, 16.3%; 605 nm, 9.5%; 650 nm, 14.2%) traps caught more individuals than other tested colors (38,45,46,53,63,116).”

L647: Or they displayed no attraction to selected semiochemicals. Alternatively, you could finish the sentence off by no known response to these semiochemicals.

We agree and have added the following sentence (lines 682–686): “It is also possible that the semiochemical lures on the traps deterred capture of Anthaxia spp. Santoiemma et al. (87) found that mean catch of Anthaxia constricticollis Bílý was significantly greater in unbaited green multiple-funnel traps than in the same traps baited with UHR ethanol plus either E,Z-fuscumol and E,Z-fuscumol acetate or racemic 3-hydroxyhexan-2-one, racemic 3-hydroxyoctan-2-one, and racemic syn-2,3-hexanediol.”

Reviewer #2:

The manuscript is very detailed and well-written, and overall, I had only minor comments (minor revision). Several appropriately chosen statistical methods were used to evaluate the results. I believe the authors have provided all relevant data in the manuscript, allowing the reviewer to gain a comprehensive understanding of the study. The English language used in the article is of a high scientific standard.

We thank the Reviewer for the positive comments.

Overall Assessment: The manuscript presents a well-structured and methodologically sound study on optimizing the detection of bark and woodboring beetles (BBWB) through the use of different trap colors and heights. The research is highly relevant for forest protection and invasive species management, providing insights with practical applications. The study's multinational scope (Canada, Poland, USA, China) and standardized methodology enhance its significance. The findings highlight how varied trap configurations improve species detection, offering valuable guidance for biosecurity and ecological monitoring programs.

Strengths:

1. Scientific and Practical Significance: The study addresses a critical aspect of invasive species monitoring, demonstrating how trap color and height influence detection efficiency.

2. Robust Experimental Design: The use of multiple funnel traps with controlled variables (color, height, lure treatment) ensures reliable data collection.

3. Comprehensive Statistical Analysis: Techniques such as GLMM, perMANOVA, and species accumulation curves strengthen the credibility of the results.

4. Broad Geographical Scope: Data from diverse regions allow for cross-site comparisons, increasing the study's applicability.

5. Potential for Practical Implementation: The findings contribute to optimizing forest pest monitoring strategies.

We thank the Reviewer for the positive comments.

Suggested Improvements (Minor Revisions):

1. Introduction

o The introduction could benefit from a clearer statement of the research question. The authors should also provide a more concise overview of the literature on trap color and height preferences of BBWB.

o Additionally, given the overall length of the manuscript, the introduction is relatively short, and the topic should be expanded further.

o Sentences like “We used….” or “We replicated…” (lines 106–113) do not fit well within the Introduction section.

In response to these three comments, we made some minor revisions to more clearly state our research goals and to clarify why we used species richness, trap catch, and detection rate as response variables.

Lines 81–83: “Placing traps in both strata should sample a broader range of BBWB species and potentially increase the number of species detected, including non-native BBWB that may be present.”

Lines 94–110: “Previous studies have tested the effects of trap color and/or trap height on the number of species richness of Buprestidae and Cerambycidae (25), Scolytinae (52), Buprestidae, Cerambycidae, and non-scolytine Curculionidae captured (34), or on mean catch abundance of individual BBWB species of BBWB (53–56), but few studies have simultaneously tested the effects of trap color and trap height on the number of species of all three taxa (Buprestidae, Cerambycidae, Scolytinae) captured (51,57). Our objectives were to evaluate the effects of trap height and trap color on species richness and abundance of BBWB species (Cerambycidae, Buprestidae, Disteniidae, and Curculionidae: Scolytinae) captured in traps, and to determine the combination(s) of trap height and color that captured the greatest number of species per trapping effort, with the overall goal of improving trapping surveillance for detection of non-native BBWB. A trapping system that increases the species richness of target taxa captured should increase the chances of detecting non-native species of related taxa that may be present (20,32). We also used mean catch per trap of individual species, or their detection rate (proportion of traps that captured at least one specimen), as response variables. Although very few relationships between trap catch and infestation level have been determined for any BBWB species [but see Hanula et al. (58)], mean Mean catch per trap and detection rate provide measures of relative efficacy among trap treatments for detecting a particular species.”

2. Ecological Justification for Trap Colors:

o While the selection of black, green, and purple traps is methodologically sound, a clearer discussion of how these colors relate to beetle visual ecology (e.g., photoreceptor spectral sensitivity, background contrast) would enhance the study’s ecological context.

We appreciate the reviewer’s comment, but we think we have provided sufficient ecological context as to why some beetles are attracted to certain colors while others are not (lines 664–678). An in-depth discussion of how and why the various beetle species respond to trap color would be interesting but is beyond the scope of this paper which one reviewer remarked was already a little long. Our paper has an applied focus on how trap color and trap height affect the community composition, species richness and abundance of bark and woodboring beetles (BBWB) captured in traps, and how that knowledge may be applied to improve trapping surveys for detection of non-native BBWB species.

3. Site-Specific Environmental Considerations:

o Differences in trap efficiency across sites may be influenced by local conditions (e.g., canopy cover, light availability, host tree composition). Including a discussion on environmental factors affecting beetle attraction would strengthen the interpretation.

We agree with the first point and we acknowledge the possible effects of site-specific differences in forest structure and tree species composition on the relative performance of the trap color-height combinations for capture of bark and woodboring beetles (lines 642–648:

“Replicating the same experimental design in North America, Europe and Asia revealed that the influence of trap colour and trap height were relatively consistent among continents for some taxa (e.g., Buprestidae: Agrilinae, Chrysochroinae; and Curculionidae: Scolytinae) but not for other taxa (Cerambycidae: Cerambycinae, Lamiinae, Lepturinae, Prioninae, Spondylidinae). This likely reflects variation in foraging habits and preferences of the BBWB species assemblages present at each site (31,33) as well as differences among sites in forest structure and tree species composition that can influence the effects of trap placement or color on BBWB trap catches (29,104–106).”

And again on lines 746–750:

“This suggests the species richness of cerambycids in different parts of the canopy is likely influenced by the species-specific foraging habits of the cerambycid species present at the survey site, as well as site-specific factors such as forest type, stand structure, and edge effects (20) which is then reflected in the species assemblages captured by the traps.”

We would prefer not to include further discussion of environmental factors that may have affected beetle catches as the manuscript is already quite long. Readers that want to explore this issue in greater depth have the option of reading the papers we have cited.

o The trap placement heights varied significantly across different countries.

This is true for the traps placed in the canopy because the height of trees varied both within and among sites, but the position of the traps relative to the canopy of each tree was relatively consistent, that is, canopy traps at all four sites were always placed in the mid to upper canopy of trees. We have added a sentence to clarify this (lines 188–191):

“Canopy traps were suspended in the mid to upper crown of dominant or co-dominant trees. The height above ground of canopy traps necessarily varied with the height of the trees in which they were placed, but their relative position was always in the mid to upper tree crown.”

Note: height of traps in the understory varied little among sites, that is, the collecting cup was 30–50 cm above the ground at all sites.

4. Clarification of Methodological Details:

o Specify the exact placement of "canopy" traps in relation to tree height and structure.

All canopy traps were placed in the mid to upper crown of trees at each site, as stated in the Methods, and now further clarified. Please see response to the comment above.

o Indicate whether sampling intervals aligned with peak beetle activity periods.

We have added the following two sentences regarding peak beetle activity periods and cite an additional 4 articles (lines 135–139):

“The trapping periods (May–September in Canada and China; May–July in USA and Poland) covered the peak flight periods of most species of BBWB at our sites (59–61). However, we likely missed the peak flight periods of some species of target taxa, e.g., the ambrosia beetles, Xylosandrus germanus (Blandford) and Xylosandrus crassisusculus (Motschulsky), for which peak flights occur in March–April (62).”

o The UHR ethanol lure should have been installed earlier, as many species of ambrosia beetles are active as early as March–April.

We agree that catches of some ambrosia beetle species would have been greater had we set the traps up earlier in the season, and we now acknowledge this on lines 135–139 in the revision (see above response).

5. Statistical and Data Reporting:

o Ensure consistency in citation of statistical methods and software (e.g., R packages used for analyses are listed in References (89) but not mentioned in the Methodology section).

We have double-checked citations and the reference lists in the original submission and the revision and confirm that all statistical methods and software (e.g., R packages) that were used and cited are listed in the References. Furthermore, we have now published the R code in the Open Government Portal and now state this (lines 349–350):

“All raw data, metadata, and R codes are available on the Government of Canada Open Government portal (102).”

o Clarify whether multiple testing corrections (e.g., FDR adjustments) were applied to pairwise comparisons.

All post-hoc tests comparing means among the six treatment combinations of trap color and trap height were done on least square means using the Tukey-Kramer post-hoc test adjusted for multiple comparisons; this controls the experiment-wise error at α = 0.05. We state this in regard to species richness on lines 249–250. However, we had failed to mention we used the same post-hoc test when comparing me

---

## [Editor Report · Decision Letter 1]

21 Mar 2025

Diversity in trap color and height increases species richness of bark and woodboring beetles detected in multiple funnel traps

PONE-D-25-03887R1

Dear Dr. Sweeney,

We’re pleased to inform you that your manuscript has been judged scientifically suitable for publication and will be formally accepted for publication once it meets all outstanding technical requirements.

Kind regards,

Ramzi Mansour

Academic Editor

PLOS ONE

---

## [Editor Report · Acceptance letter]

PONE-D-25-03887R1

PLOS ONE

Dear Dr. Sweeney,

I'm pleased to inform you that your manuscript has been deemed suitable for publication in PLOS ONE. Congratulations! Your manuscript is now being handed over to our production team.

Kind regards,

on behalf of

Dr. Ramzi Mansour

Academic Editor

PLOS ONE